EMBO
Molecular Medicine

# Navigating from cellular phenotypic screen to clinical candidate: selective targeting of the NLRP3 inflammasome

Rosalie Matico[1,13], Karolien Grauwen[2,13], Dhruv Chauhan[2,13], Xiaodi Yu [1,13], Irini Abdiaj[3], Suraj Adhikary[1], Ine Adriaensen[4], Garcia Molina Aranzazu[3], Jesus Alcázar[3], Michela Bassi[5], Ellen Brisse [2], Santiago Cañellas [3], Shubhra Chaudhuri[6], Francisca Delgado[3], Alejandro Diéguez-Vázquez[3], Marc Du Jardin[7], Victoria Eastham[1], Michael Finley [1], Tom Jacobs[8], Ken Keustermans[9], Robert Kuhn[10], Josep Llaveria [3], Jos Leenaerts[5], Maria Lourdes Linares[3], Maria Luz Martín[3], Rosa Martín-Pérez [2], Carlos Martínez[3], Robyn Miller[1], Frances M Muñoz[1], Michael E Muratore[5], Amber Nooyens[2], Laura Perez-Benito[11], Mathieu Perrier[5], Beth Pietrak[1], Jef Serré [2], Sujata Sharma[1], Marijke Somers[12], Javier Suarez[1], Gary Tresadern[11], Andres A Trabanco [3], Dries Van den Bulck[1], Michiel Van Gool[5], Filip Van Hauwermeiren[2], Teena Varghese [1], Juan Antonio Vega[3], Sameh A Youssef[8], Matthew J Edwards[2], Daniel Oehlrich[5] & Nina Van Opdenbosch [2]✉

## Abstract

The NLRP3 inflammasome plays a pivotal role in host defense and drives inflammation against microbial threats, crystals, and danger-associated molecular patterns (DAMPs). Dysregulation of NLRP3 activity is associated with various human diseases, making it an attractive therapeutic target. Patients with *NLRP3* mutations suffer from Cryopyrin-Associated Periodic Syndrome (CAPS) emphasizing the clinical significance of modulating NLRP3. In this study, we present the identification of a novel chemical class exhibiting selective and potent inhibition of the NLRP3 inflammasome. Through a comprehensive structure–activity relationship (SAR) campaign, we optimized the lead molecule, compound A, for in vivo applications. Extensive in vitro and in vivo characterization of compound A confirmed the high selectivity and potency positioning compound A as a promising clinical candidate for diseases associated with aberrant NLRP3 activity. This research contributes to the ongoing efforts in developing targeted therapies for conditions involving NLRP3-mediated inflammation, opening avenues for further preclinical and clinical investigations.

**Keywords** Novel Inhibitor; NLRP3; IL-1β; Inflammasome; Clinical Candidate

**Subject Categories** Immunology; Pharmacology & Drug Discovery

## Introduction

Dysregulation of the NLR-family pyrin domain containing 3 (NLRP3) inflammasome has emerged as a key contributor to the pathogenesis of numerous human diseases including atherosclerosis (Duewell et al, 2010), gout (Martinon et al, 2006), multiple sclerosis (Inoue et al, 2012), Alzheimer's disease (Halle et al, 2008), and several cancers (Kolb et al, 2014; Karki et al, 2017; Sekaran et al, 2024). The NLRP3 inflammasome is a cytosolic, multiprotein complex consisting of a pattern recognition receptor (PRR), an apoptosis-associated speck-like protein containing a caspase recruitment domain (ASC), and a cysteine protease pro-caspase-1. Aberrant activation of NLRP3, illustrated by Cryopyrin-Associated Periodic Syndromes (CAPS) caused by genetic mutations, underscores the importance of understanding and modulating its function.

A wide variety of damage-associated molecular patterns (DAMPs) and pathogen-associated molecular patterns (PAMPs) activate the

[1]Janssen Research & Development, LLC, Discovery Technologies and Molecular Pharmacology (DTMP), Spring House, PA 19044, USA. [2]Janssen Interventional Oncology, Turnhoutseweg 30, 2340 Beerse, Belgium. [3]Janssen Research & Development, LLC, Global Discovery Chemistry (GDC), C. Río Jarama, 75, 45007 Toledo, Spain. [4]Janssen Research & Development, LLC, In Vivo Sciences (IVS), Turnhoutseweg 30, 2340 Beerse, Belgium. [5]Janssen Research & Development, LLC, Global Discovery Chemistry (GDC), Turnhoutseweg 30, 2340 Beerse, Belgium. [6]Janssen Research & Development, LLC, Preclinical Sciences and Translational Safety (PSTS), Spring House, PA 19044, USA. [7]Janssen Research & Development, LLC, Discovery Pharmaceutics, Turnhoutseweg 30, 2340 Beerse, Belgium. [8]Janssen Research & Development, LLC, Preclinical Sciences and Translational Safety (PSTS), Turnhoutseweg 30, 2340 Beerse, Belgium. [9]Charles River Laboratories, Turnhoutseweg 30, 2340 Beerse, Belgium. [10]Janssen Interventional Oncology, Spring House, PA 19044, USA. [11]Janssen Research & Development, LLC, Therapeutic Discovery, Turnhoutseweg 30, 2340 Beerse, Belgium. [12]Janssen Research & Development, LLC, Drug Metabolism and Phamacokinetcs (DMPK), Turnhoutseweg 30, 2340 Beerse, Belgium. [13]These authors contributed equally: Rosalie Matico, Karolien Grauwen, Dhruv Chauhan, Xiaodi Yu. ✉E-mail: nvanopde@its.jnj.com

sensor NLRP3, resulting in its conformational change allowing recruitment of ASC (Yu et al, 2024). Consequently, oligomerization of NLRP3 and ASC leads to the recruitment of pro-caspase-1 to ASC and proximity-induced autoproteolysis resulting in caspase-1 activation (Broz et al, 2010). Distinctive to NLRP3 is its two-step activation pathway, requiring NFκB-mediated transcriptional upregulation and post-translational modifications during the 'priming' step, followed by a secondary signal leading to inflammasome activation (Bauernfeind et al, 2009; O'Keefe et al, 2024). Downstream, the pro-inflammatory cytokines pro-interleukin 1β (IL-1β) and pro-interleukin 18 (IL-18) are cleaved into their bio-active forms by activated caspase-1. Simultaneously, caspase-1-dependent gasdermin D (GSDMD)-cleavage results in the release of its N-terminal fragment that oligomerizes to form pores in the plasma membrane resulting in pyroptosis—a lytic, inflammatory form of cell death (Lamkanfi and Dixit, 2012).

The exploration of inflammasomes extends beyond NLRP3 to include four distinct, extensively studied members: NLRP1, NLRC4, PYRIN, and AIM2, each characterized by unique sets of specific agonists. NLRP1 was the first inflammasome identified as a caspase-activating complex (Martinon et al, 2002). Subsequent investigations revealed that the murine NLRP1 isoform undergoes activation through a proteolysis event induced by *Bacillus anthracis* Lethal Toxin (LeTx) and human NLRP1 by proteases from diverse picornaviruses (Boyden and Dietrich, 2006; Tsu et al, 2021). Distinct activation mechanisms are attributed to other inflammasomes. NLRC4 responds to bacterial flagellin (FlaTox) and proteins of the type III secretion system (NdlTox), achieved through direct interaction with the NAIP receptors. The PYRIN inflammasome, on the other hand, is triggered by bacterial toxins that inactivate RhoA GTPases (Mariathasan et al, 2004). In addition, AIM2 functions as a sensor for altered and mislocalized, intracellular and cytoplasmic double-stranded DNA molecules, culminating in inflammasome activation and initiation of an innate immune response (Lugrin and Martinon, 2018). This diversity in inflammasome activation mechanisms underscored the sophisticated nature of the host immune system's response to various pathogenic stimuli. Therefore, a deeper understanding of their roles in health and disease is required, which can offer potential targets for therapeutic interventions in a spectrum of inflammatory diseases.

While the pivotal role of NLRP3 in diverse pathologies is well-established, therapeutic intervention targeting this inflammasome has gained momentum. Notably, in 2015, the small-molecule inhibitor MCC950 (CRID3 or CP-456773) showcased potent and selective NLRP3 inhibition in several preclinical models and human samples from CAPS patients, providing an initial breakthrough (Coll et al, 2015). MCC950 is part of the family of diaryl sulfonylurea-containing compounds that was originally identified as novel IL-1β processing inhibitors (Lamkanfi et al, 2009). Confirmation of MCC950 direct binding to the Walker A motif of the NLRP3-NACHT domain identified a novel mode of action (MOA) for NLRP3 inhibition (Vande Walle et al, 2019). Since then, several other small-molecule inhibitors have been reported to target the NLRP3 inflammasome with some of these molecules currently in clinical development (Charan et al, 2023; Zhang et al, 2023; Vande Walle and Lamkanfi, 2024). Several compounds show similar MOA to MCC950 through a direct interaction with NLRP3. First-line derivatives of MCC950, Inzomelid and Somalix (Roche), completed a Phase I trial but have not entered a Phase II study to date (Marino, 2023). NodThera's NT-0796 is brain-penetrant and recently progressed to Phase Ib/IIa in

Parkinson's disease (Coll et al, 2022). Nonetheless, concerns over hepatotoxicity within the diaryl sulfonylurea compound class have spurred the exploration of alternative pharmacophores (Charan et al, 2023; Shah et al, 2015; Mangan et al, 2018). In addition to MCC950 analogs other structurally different clinical candidates act as pathway inhibitors instead of on-target NLRP3 inhibitors. Among these structurally different small molecules, the most clinically advanced, which directly targets the NLRP3 pathway is RRx-001 that has now entered phase III study in patients with small cell lung cancer (SCLC). In addition, RRx-001 can cross the blood-brain barrier, and is under investigation for neurodegenerative diseases (Chen et al, 2021; Jayabalan et al, 2023; Oronsky et al, 2023). Recently, OLT1177 (Olatec Therapeutics) completed a phase II trial for osteoarthritis and also being evaluated for Alzheimer's disease, arthritis or rare auto-inflammatory disease (Marchetti et al, 2018; Sánchez-Fernández et al, 2019). The old anti-allergic, clinical drug, Tranilast (Rizaben®), was identified as a direct NLRP3 inhibitor and is under clinical evaluation, however it is likely to display off-target activity on other inflammatory pathways (Huang et al, 2018; Matsumura et al, 2022). CY-09 directly inhibits the ATPase activity of NLRP3 and shows protection against inflammation in animal models (Jiang et al, 2017). Alternatively, the compound SLC-3037 has been identified as a potential inhibitor of the NLRP3 inflammasome. Its mechanism of action involves disrupting the binding of NEK7 to NLRP3, presenting new opportunities for inhibiting the NLRP3 inflammasome (Park et al, 2023). A recent review provides a thorough overview of the status of NLRP3 inhibitors, eloquently detailing the characteristics of all the aforementioned drug candidates (Vande Walle and Lamkanfi, 2024).

In this study, we focus on a phenotypic high-throughput screening campaign that identified a novel, differentiated small-molecule NLRP3 inhibitor, compound A, characterized by its high potency and selectivity over the other inflammasomes and NFκB pathway. Target engagement to NLRP3 was confirmed using biochemical assays and Cryogenic Electron Microscopy (Cryo-EM). This novel compound presents a promising avenue for the development of NLRP3-selective small molecules inhibitors, offering potential therapeutic benefits distinct from the conventional diaryl sulfonylurea-containing compounds. As we unravel the unique features of compound A, our research paves the way for a new era in NLRP3-targeted therapeutic strategies.

## Results

### Phenotypic screening identifies novel chemistry class that directly interacts with human NLRP3

The screening assay was performed using a murine macrophage cell line (J774A.1) primed with lipopolysaccharide (LPS) for 2 h followed by treatment with nigericin (Nig) to activate the NLRP3 inflammasome. Cells were treated with the cell impermeant dye Sytox green, which is only taken up by cells with a compromised cell membrane. The assay window was compared between unstimulated and Nig-stimulated cells. Compounds that inhibit the inflammasome formation, block pyroptosis, thereby preventing uptake of the Sytox green dye. Conversely, cytotoxic compounds, which led to cell death were therefore excluded from the list of potential hits during the screening process. Using this phenotypic cellular assay, we roughly screened a million compounds (assay window S/B: 14, variability factor: Z' 0.8)

and selected ~14,000 compounds as hits based on inhibition activity >60% normalized to the inhibitory activity of MCC950. These hits were subsequently confirmed in the primary NLRP3 assay and counter-screened in an assay using the NLRC4-selective inflammasome activator FlaTox to induce pyroptosis. Compounds falling in the lower right quadrant (>60% NLRP3 and <30% NLRC4) were identified as confirmed hits (Fig. 1A). Within this quadrant, we identified 27 structurally distinct chemical classes, which were then further triaged using cell death and cytokine readouts (Appendix Fig. S1). This triage and the use of in-house promiscuity data indicated that the majority of the identified clusters were either false positives or pathway inhibitors, which was an undesired MOA, leading to the selection of cluster 1. Within the identified cluster 1, compound B was used to demonstrate the appropriate phenotypes on cell death and IL-1β-induced by Nig in LPS-primed J774A.1 cells, measured by a reduction in Sytox green-positive cells (Fig. 1B) and reduced IL-1β release (Fig. 1C). Using the FlaTox-mediated NLRC4 counter screen, we showed selective inhibition for NLRP3 by compound B in the murine J774A.1 macrophage cell line (Fig. 1D). Next in the screening funnel, we stimulated human peripheral blood mononuclear cells (PBMCs) with LPS for 6 h looking to identify potent and selective inhibitors of IL-1β while the general NF-κB-mediated IL-6 and TNF remained unaffected. Interestingly, we found a potent, dose-dependent inhibition of NLRP3 inflammasome-induced IL-1β release by compound B, but not the general IL-6 and TNF cytokines, consistent with its specificity for inhibition of NLRP3 inflammasome pathway (Fig. 1E). Cluster 1 consists of compound B, C, and D, which are tricyclic core compounds with a side chain amide belonging to the pyrolo-triazine acetamide compound class (Fig. 1F). To this end, cluster 1 will be represented by compound C *vide infra* (Fig. 1F), which showed a 0.041 µM potency on IL-1β, while IL-6 and TNF were not inhibited below 10 µM (Fig. 1G) in human PBMCs. Compound D was identified as a structural homolog of compound C however was unable to block the NLRP3 inflammasome (Fig. 1F,G). Next, target engagement was investigated by using recombinant human NLRP3 with compounds B, C, D, and MCC950 as a reference compound. Nano-differential scanning fluorometry (DSF) with recombinant MBP-hNLRP3-ΔPYD protein demonstrated a shift in melting temperature when co-incubated with MCC950, compound B and C indicative of protein stabilization due to compound interaction while there was no thermal shift observed with compound D (Appendix Fig. S2). Addition of adenosine diphosphate (ADP) to the recombinant NLRP3 led to an increased stabilization of all three active compounds tested (but not inactive compound D), potentially making a more rigid confirmation of NLRP3. All together, we identified cluster 1 as a potent, selective, chemically distinct pharmacophore directly interacting with NLRP3 to prevent its activation. During the preparation of this manuscript, the same chemical class was described as potent and selective NLRP3 inhibitors, thereby confirming our screening campaign (Vande Walle et al, 2024; Velcicky et al, 2024).

## Cryo-EM structure of MBP-hNLRP3-ΔPYD complex shows that compound C locks NLRP3 in a closed state thereby inhibiting the ATPase activity

The NLRP3 protein consists of a N-terminal pyrin (PYD) domain, a C-terminal leucine-rich repeat (LRR) domain and central a functional NACHT domain containing a nucleotide-binding

domain (NBD), helical domain 1 (HD1), a winged helix domain (WHD) and helical domain 2 (HD2) (Yu et al, 2024) (Fig. EV1A). Small molecules like MCC950 inhibit NLRP3 activation by targeting the NACHT domain and locking it in a closed conformation (Dekker et al, 2021; Hochheiser et al, 2022; Ohto et al, 2022; McBride et al, 2022; Velcicky et al, 2024; Yu et al, 2024; Ohba et al, 2023) (Fig. EV1B–D). After establishing the specificity of compound C for NLRP3, the cryo-EM structure of MBP-hNLRP3-ΔPYD in complex with ADP and compound C was assessed. It formed a tetramer via the MBP-NACHT and back–back (between LRR and LRR) interactions with D2 symmetry (Fig. 2A; Appendix Figs. S3 and S4A–E). The compound C bound MBP-hNLRP3-ΔPYD and the closed NLRP3 (PDBID: 7VTP) protomer structures can be overlayed with a root-mean-square deviation (RMSD) 2.05 Å at the NACHT-LRR region indicating that MBP-hNLRP3-ΔPYD adopts a closed pose in presence of compound C similar to MCC950 (Figs. 2B–D and EV1). Two non-protein densities were observed in the NACHT core with the Walker A motif situated in between, and modeled as ADP, and compound C, respectively (Fig. 2E; Appendix Fig. S4F). This proximity supports the improved nanoDSF stabilization of compound C in the presence of ADP. A comparison of the interactions of compound C and MCC950 with NLRP3 revealed that both compounds bind to the same pocket however interact with the protein differently (Yu et al, 2024) (Fig. 2E). The tricyclic moieties of MCC950 and compound C are deeply embedded in a hydrophobic pocket, while the amide moiety linker of each compound is anchored by different amino acid residues. MCC950's linker is anchored by the Walker A motif and Arg578 with Arg351, while compound C's linker is anchored by the Walker A motif and Arg578 with Asp662, placed further away from the Walker A motif. The isopropyl furan moiety of MCC950 or pyrimidine moiety of compound C points outward towards the LRR region. In addition, the binding of these compounds causes changes in the side chains of Phe575, Glu629, and Met661 (Fig. 2E). Hydrogen–deuterium exchange (HDx) mass spectrometry analysis revealed Walker A&B motifs showed reduced deuterium uptake upon compound C binding (Appendix Fig. S5A–C), suggesting the binding of compound C could perturb the ATPase hydrolysis activity of the NACHT domain. Indeed, an ATPase hydrolysis assay showed that compound C inhibits the ATPase hydrolysis activity of MBP-hNLRP3-ΔPYD similar to reference compound MCC950 (Appendix Fig. S6A–C). Together, these data indicate that compound C locks NLRP3 in the closed state by interacting with all the NACHT subdomains and inhibiting the ATP hydrolysis necessary for NLRP3 activation.

## A structure–activity relationship campaign evolved the hit series toward a clinical candidate

After a thorough evaluation of the high-throughput screening (HTS) hit series, we identified an initial lead with compound C within the chemotype, the profile of which is representative for the series. The series displayed good potency with many examples <100 nM, but unfortunately, they all suffered from significant microsomal instability in mouse and human as seen by $CL_{int}$ (intrinsic clearance) human/mouse of 53/324 (µL/min/mg P), respectively (Fig. 3A).

Based on this observation, the initial focus was combating the metabolic instability found in cluster 1 to enable a Proof-of-

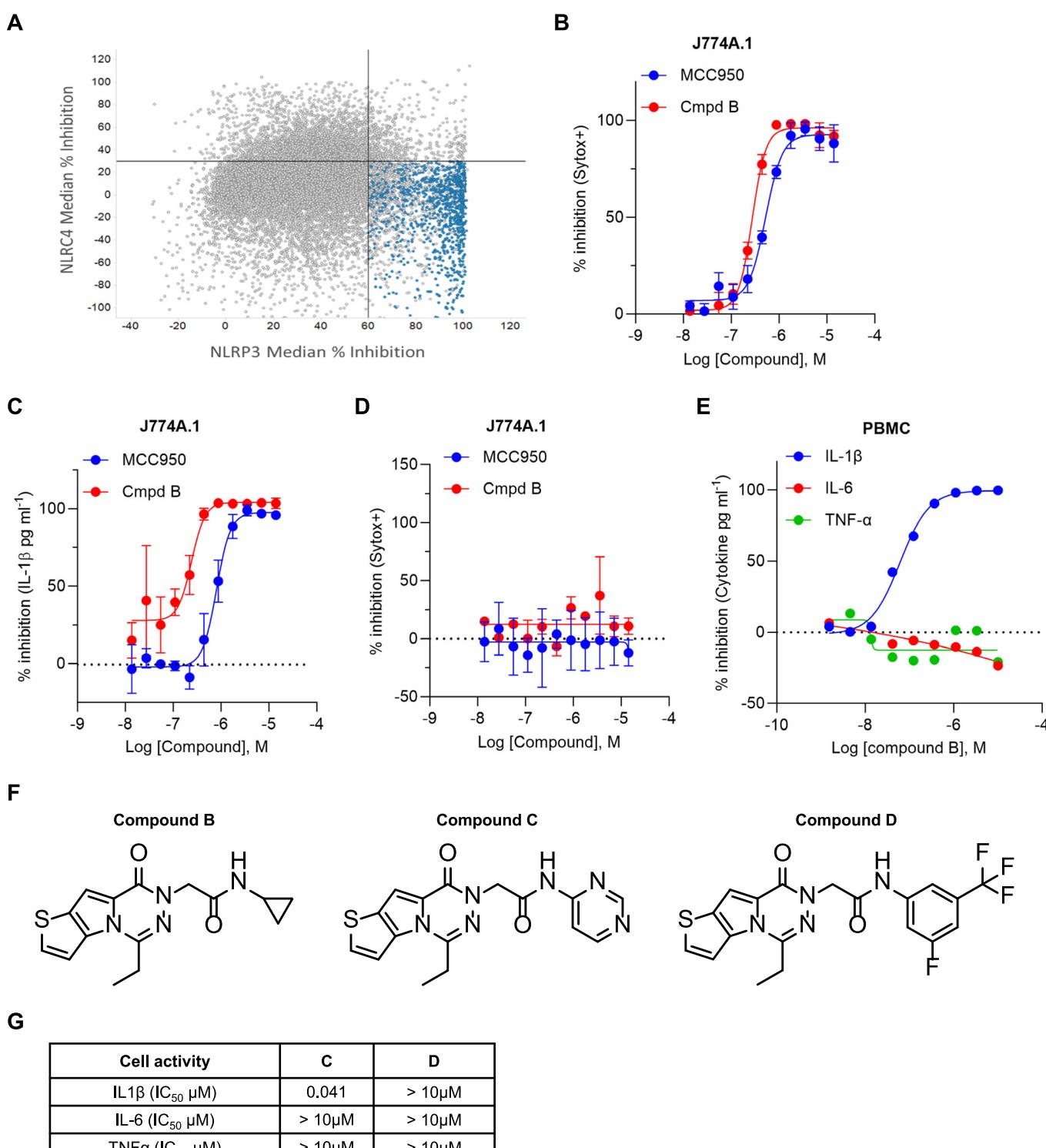

| Cell activity | C | D |
|---|---|---|
| IL1β ($IC_{50}$ μM) | 0.041 | > 10μM |
| IL-6 ($IC_{50}$ μM) | > 10μM | > 10μM |
| TNFα ($IC_{50}$ μM) | > 10μM | > 10μM |

Concept (PoC) in vivo study using LPS-induced NLRP3 activation in C57BL/6 mice and benchmarking against MCC950. The first compound to demonstrate sufficient metabolic stability for in vivo PoC was compound E. Due to the limited formalizability caused by the low kinetic solubility at pH 7.4 being <5 μM, compound E was consequently dosed using PEG400 as formulation (Fig. 3A).

Although compound E had limited solubility, it was dosed orally at 50, 25, and 12.5 mg/kg and still showed a dose-dependent inhibition of IL-1β in the blood in LPS-induced inflammation (in vivo) model (Fig. 3B). Compound E demonstrated a significant inhibition of IL-1β, which led to increased confidence in our approach and series, and resulted in efforts to diversify the

**Figure 1.   Identification of a novel chemistry class of NLRP3 inhibitors.**

(**A**) Cell death was determined using J774A.1 cells treated with LPS + Nig (NLRP3) or FlaTox (NLRC4) in the presence of 14 µM test compound and 1 µM Sytox green to identify NLRP3-specific inhibitors. The fourth quadrant was classified as >60% inhibition on NLRP3 and <30% inhibition on NLRC4. Normalized data was used for hit selection. Raw data ($n = 3$ for each compound) was converted to % inhibition (% INH) in Genedata Screener using the Percent of Control (generic) method. The normalization process included two key parameters: the Central Reference, which was the Neutral Control (DMSO), and the Scale Reference was established as the Inhibitor Control (MCC950). (**B**) Cell death was determined using LPS-primed J774A.1 cells treated with a dose response of compound B and reference MCC950 followed by treatment with 20 µM Nig (2 h) using 1 µM Sytox green. Plates were read using a PHERAstar FSX (BMG Labtech) using 487 nm wavelength. Dose response was performed in duplicate, or triplicate and data is depicted in the graph as mean $+/-$ SD of two independent repeats ($n = 2$). (**C**) Levels of IL-1β were determined by means of AlphaLISA technology using LPS-primed J774A.1 cells treated with a dose response of compound B and reference MCC950 followed by treatment with 20 µM Nig (1 h). Dose response was performed in duplicate, or triplicate and data is depicted in the graph as mean $+/-$ SD of two independent repeats ($n = 2$). (**D**) Cell death was determined using J774A.1 cells treated with a dose response of compound B and reference MCC950 followed by treatment with 500 ng/ml FlaTox using 1 µM Sytox green. Plates were read using a PHERAstar FSX (BMG Labtech) using 487 nm wavelength. Dose response was performed in duplicate, or triplicate and data is depicted in the graph as mean $+/-$ SD of two independent repeats ($n = 2$). (**E**) The inhibition of IL-1β, IL-6 and TNF by compound B and MCC950 on 100 ng/ml LPS-treated human PBMCs (6 h) was determined using Mesoscale discovery (MSD). PBMCs were used in two independent experiments ($n = 2$) each in dose response and representative data is depicted in the graph. (**F**) Structure of representative compounds B, C, and D identified from the screening funnel as selective NLRP3 inhibitor (compound B and C) and inactive homolog (compound D). (**G**) IC$_{50}$ for IL-1β, IL-6, and TNF was determined using 100 ng/ml LPS-stimulated PBMCs (from healthy donors) in the presence of compounds C and D. PBMCs were used in two independent experiments ($n = 2$) each in dose response and pooled data used to calculate the IC$_{50}$. Source data are available online for this figure.

chemical space, resulting in several novel chemotypes, with the aim of identifying profiles with increased metabolic stability and solubility (Figs. 3B and EV2A,B).

The concern of the suboptimal compound profile led us to consider the impact of the tricyclic core, and search for smaller heterocyclic replacements. Indeed, some of the properties, including metabolic stability (CL$_{int}$) and solubility, could be modulated by the tricyclic core structure in examples C and E (Velcicky et al, 2024; Li et al, 2023) (Fig. 3A). Initial attempts to truncate the core were highly successful with the identification of several structurally distinct clusters. The core of most interest was the phthalazine, represented by compound G and H (Fig. 3C). During the optimization of this chemotype, we identified the need to substitute the 6-position with a group to mimic the third ring of the tricyclic system to optimally engage the pocket. Interestingly, the optimal moieties at the 6-position of the phthaliazine were lipophilic in character, of which the trifluoromethyl-substitution displayed the best potency combined with metabolic stability. Initially, we profiled compound F, which demonstrated less than ideal potency (IC$_{50}$ 176 nM), suffered from low solubility, and low metabolic stability in the mouse (Fig. 3C). During the optimization, it was identified that improved potency and stability could be achieved by exchanging the pyrimidine amide on the right-hand-side (RHS) of these systems with bicyclic amides, as illustrated with the 1,2,4-triazolo-pyridine analog compound G and the 1,2,4-triazolo-pyridazine compound H (Fig. 3C). Both analogs compound G and H demonstrated sub-100 nM activity against NLRP3, good metabolic stability in both species with extraction ratios of <7.7 in humans and 12.6 and 13.1, respectively, in the mouse (Fig. 3C). Compound H also exhibited improved permeability of 22.6 over compound F and G, with 9.2 and 14.0 respectively. However, it did not show increased solubility over compound G (Fig. 3C).

Consequently, further optimization was necessary, and efforts were focused on combining the selected northern hemisphere with alternatives to the isopropyl. The ideal combination was identified by substituting the isopropyl with a dimethyl-amine which led to similar potency (67 nM) and metabolic stability of 7.7 and 12.3 for human and mouse, respectively. This came with improved solubility at both pH 2 and 7.4 of 64.3 and 68.4 µM respectively

(Fig. 3D). This compound, hereafter compound A was progressed in our flow chart towards in vivo target engagement. Compound A was dosed in a mouse PK experiment, to ensure that the in vitro profile translated in vivo and demonstrated low clearance with 6.21 mL/min/Kg which translated into a reasonable half-life (T½) of almost 1 h, and a very good bioavailability (F%), with greater than 3 µM plasma levels being measured 4 h post dosing (Fig. 3E). Taken together, compound A was identified as a potential lead with the desired properties and was further evaluated in vitro and in vivo.

## Compound A targets the same NACHT binding pocket as MCC950 or compound C

Compound A shares key features with compound C however differs with a dicyclic head and tail instead of a tricyclic head. To explore compound A's binding mode, molecular dynamics (MD) simulations were conducted, using compound C's binding to NLRP3 as a template. The simulations used a monomeric NLRP3 bound to compound C, excluding the oligomer formation and ATP/ADP. Three MD simulations were run: one with NLRP3 alone (Apo), and two with NLRP3 bound to compound C and compound A, respectively. The MD simulations revealed that all three conditions adopted a closed conformation within 10 ns, regardless of compound binding. The measured distances between NBD-Val353 and HD2-Glu629 were consistent with closed NLRP3 structures and much smaller than those in the activated NLRP3 inflammasome (Fig. EV1F). In addition, distances between the compounds and NLRP3 were measured at two critical sites: Arg578 and Glu636. Compound C showed closer interaction with Arg578, while compound A formed more stable interactions with Glu636, likely due to an amine group replacing the ethyl group in compound C (Fig. EV1B,C,F).

Overall, the simulation results suggest that compound A likely targets the same NACHT binding pocket as compound C, with a similar binding mode. Moreover, compound A exhibits a larger interaction area with NLRP3 compared to other small molecules targeting this pocket. These findings indicate that compound A may be more effective in stabilizing NLRP3 in its closed conformation, thus inhibiting its activation.

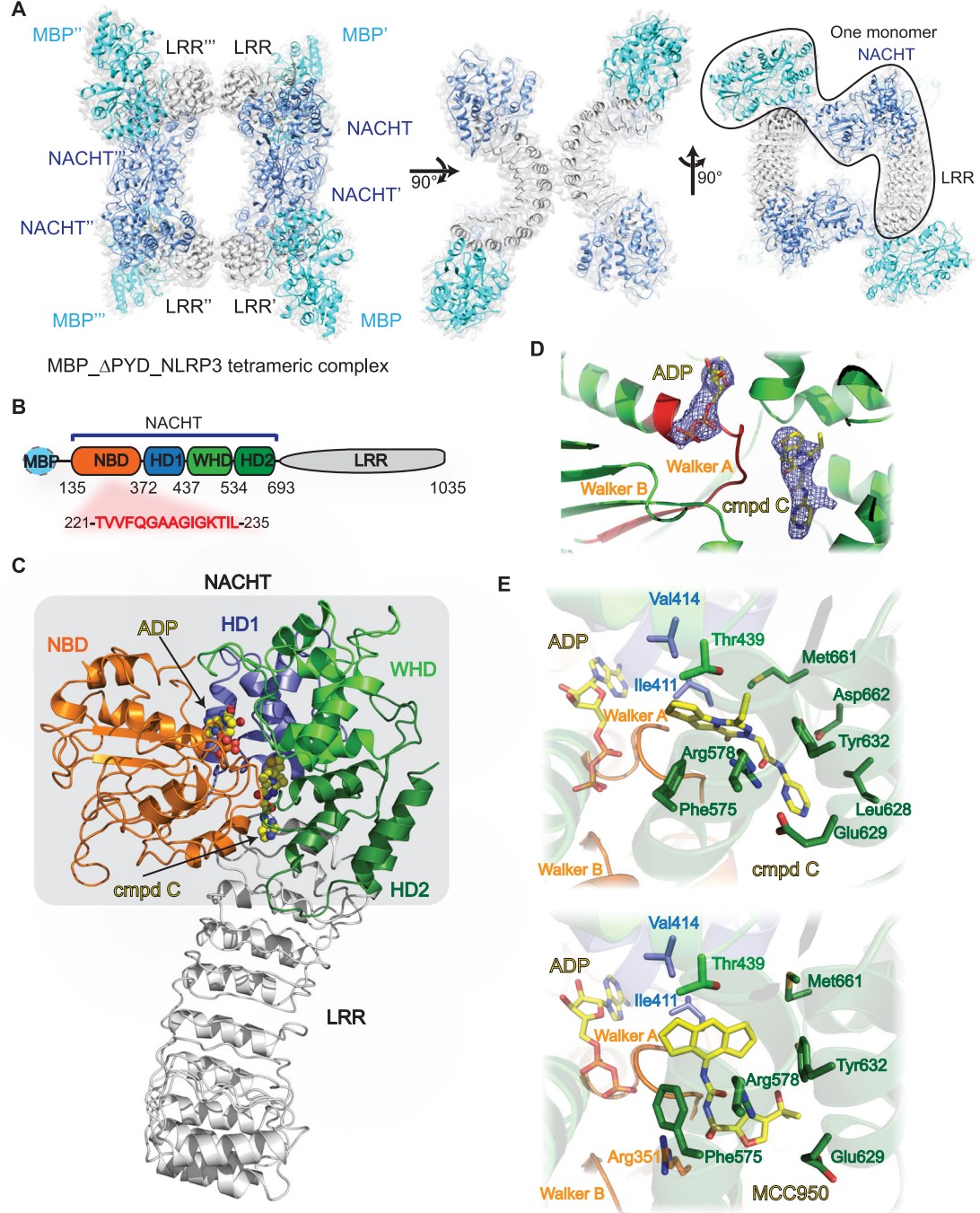

**Figure 2. Cryo-EM confirmed target engagement by direct interaction with human NLRP3 protein.**

(A) Different views of Cryo-EM map of NLRP3 complex with compound C. The MBP, NACHT, and LRR were colored in cyan, slate, and gray, respectively. One monomer of NLRP3 tetramer complex was highlighted. (B) Domain architecture of NLRP3. (C) Chemical structure of compound C. (D) The structure NLRP3 complex with compound C was color-coded as Fig. 2B, and depicted in cartoon form, while ADP and compound C were represented as ball-and-stick models. (E) Close views of compound C (top), and MCC950 (bottom) binding sites. NLRP3 was shown as cartoon and followed the color codes as (B). The ligands and the side chains of key residues were shown as sticks. Walker A and B motifs were highlighted. Source data are available online for this figure.

## Novel compound A is a highly selective and potent inhibitor of the NLRP3 inflammasome

Next, to assess the selectivity of compound A, we utilized a panel of known triggers to examine its inhibitory effect on the NLRP3, NLRC4,

Pyrin, AIM2, and NLRP1b inflammasomes. LPS-primed bone marrow-derived macrophages (BMDMs) from wild-type (WT) C57BL/6N mice were pretreated with 10 μM compound A followed by stimulation with Nig and ATP (for NLRP3), FlaTox (for NLRC4), TcdA (for Pyrin), and dsDNA transfection (for AIM2). In NLRP3-

**A**

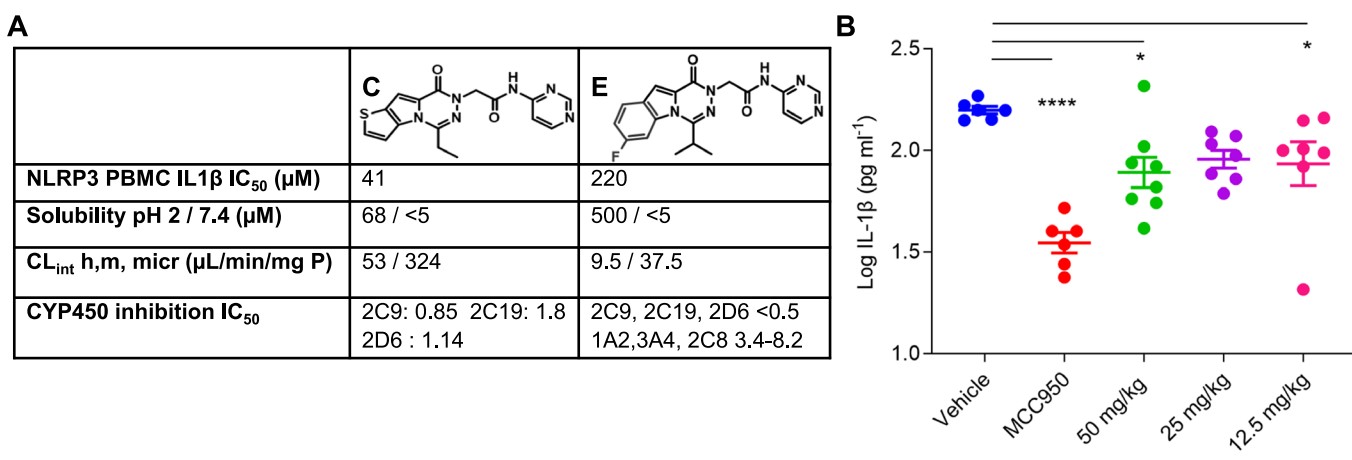

| | **C** | **E** |
|---|---|---|
| NLRP3 PBMC IL1β IC$_{50}$ (µM) | 41 | 220 |
| Solubility pH 2 / 7.4 (µM) | 68 / <5 | 500 / <5 |
| CL$_{int}$ h,m, micr (µL/min/mg P) | 53 / 324 | 9.5 / 37.5 |
| CYP450 inhibition IC$_{50}$ | 2C9: 0.85  2C19: 1.8  2D6 : 1.14 | 2C9, 2C19, 2D6 <0.5  1A2,3A4, 2C8 3.4-8.2 |

**B**

**C**

| | **F** | **G** | **H** |
|---|---|---|---|
| NLRP3 PBMC IL1β IC$_{50}$ (µM) | 176 | 0.072 | 0.063 |
| Solubility pH 2 / 7.4 (µM) | 4.5 / 0.6 | 205 / 15 | 9.3 / 6.6 |
| CL$_{int}$ h,m, micr (µL/min/mg P) | 8 / 93 | <7.7 / 12.6 | <7.7 / 13.1 |
| MDCK  A to B (+GF) | 9.2 | 14 | 22.6 |
| Efflux Ratio | 1.3 | >51 | 47 |

**D**

| | **A** |
|---|---|
| NLRP3 PBMC  IL1β IC$_{50}$ (µM) | 0.067 |
| Molecular weight (g/mol) | 432.4 |
| ChromLogD pH 2.6/7.4/10.5 | 2.04 / 1.91 / 2.06 |
| TPSA | 108 |
| Solubility pH 2 / 7.4 (µM) | 64.3 / 68.4 |
| CL$_{int}$ h,m, micr (µL/min/mg P) | <7.7 / 12.3 |
| MDCK  A to B (+GF) | 26.8 |
| Efflux Ratio | >110 |

**E**

| | **A** |
|---|---|
| Cl,pl mL/min/Kg | 6.21 |
| Bioavailability F % | 111 |
| Half-life T½  (h) | 0.96 |
| Volume of distribution at steady state (Vdss (L/Kg)) (TBW) | 0.236 (1.32x) |
| 4hr [total] / [free] 5 mg/kg pk  PO nM PK | 3,171 / 69.7 |

**F**

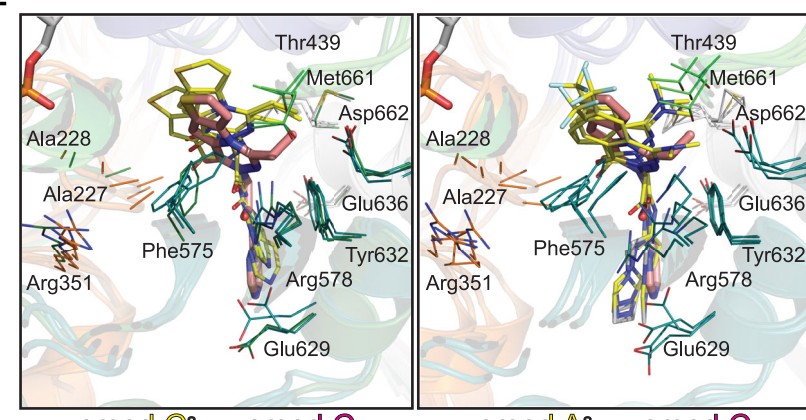

cmpd C$^s$ vs cmpd C        cmpd A$^s$ vs cmpd C

◄ **Figure 3. Structure–activity relationship campaign led to the identification of a lead clinical candidate.**

(A) Phys-chem characteristics of tricycle hits identified from HTS. (B) In vivo PoC using compound E in the LPS-induced NLRP3 activation model in C57BL/6 mice. Per group, 8 animals are used and mean +/− SEM is depicted for IL-1β measured by ELISA. One-way ANOVA with Dunnett's multiple comparison test was performed: MCC950 group: ****$P < 0.0001$, 50 mg/kg group: *$P = 0.0154$, 12.5 mg/kg group: *$P = 0.0491$. (C) Phys-chem characterization of different bicycle compounds. (D) Phys-chem characterization of the clinical lead compound. (E) Pharmacokinetic characteristics of the clinical lead compound identified in mice. Testing was performed in vivo with 5 mg/kg PO/1 mg/kg IV. (F) Structural comparison of stimulated(s) compounds C (left) and A (right) with the cryo-EM structure of the NLRP3 complex bound to compound C. Stimulated compounds are shown in yellow, while the compound C from the cryo-EM structure is colored pink. Only three stimulation frames (50, 500, and 1000) are displayed as representatives. Small molecules are depicted as sticks, with NLRP3 shown as a cartoon and color-coded as in EV1A. Key interacting residue side chains are represented as lines with colors indicating their respective subdomains. Abbreviations: Inhibitory concentration 50 (IC$_{50}$), intrinsic clearance based on microsomal incubations (CL$_{int}$), cytochrome P450 (CYP450), Madin–Darby canine kidney (MDCK) cells, Topological Polar Surface Area (TPSA), volume of distribution at steady state (Vdss). Source data are available online for this figure.

stimulated conditions, both cell death and IL-1β levels were strongly reduced, while compound A exhibited minimal effects on NLRC4, Pyrin, and AIM2 inflammasomes (Fig. 4A,B). Assessment of NFκB priming, through LPS and Pam3CSK4 stimulation with or without pretreatment of compound A, showed no impact on IL-6 and TNF cytokine levels, and no cellular cytotoxicity was observed after 16 h under the same conditions (Fig. EV2C–E). Furthermore, all established markers of inflammasome activation such as cleavage of caspase-1, GSDMD, and IL-1β were examined. Similar to MCC950, treatment of BMDMs with 10 µM compound A, blocked the cleavage of all three markers in Nig-treated but not in FlaTox-treated condition, indicating selectivity for NLRP3 inhibition over NLRC4 (Fig. 4C).

Moreover, stimulation of LPS-primed BMDMs from BALB/c mice with compound A show complete inhibition of NLRP3, while revealed no inhibition of the NLRP1b (LeTx) as well as NLRC4 (FlaTox) inflammasome, as evidenced by no marked effect on IL-1β release and cell death induction (Fig. 4D,E). Detection of ASC specks, another well-defined feature of inflammasome biology, was assessed in NLRP3 versus NLRC4-stimulated conditions using ASC oligomerization through DSS-crosslinking. Similar to MCC950, compound A resulted in the inhibition of ASC oligomerisation only upon Nig treatment while NLRC4-induced ASC oligomers were unaffected (Fig. EV2F). Analogously, immunofluorescence staining in BMDMs showed a dose-dependent inhibition of Nig-induced ASC specks by compound A (Fig. 4F). Furthermore, we tested compound A specificity for NLRP3 inhibition in human ASC-mCherry-expressing THP-1 cells. Again, compound A inhibited ASC speck formation in a dose–response manner in Nig-treated cells however had no effect on ASC specks formed by NLRC4 activation in NeedleTox (NdlTox)-treated cells (Appendix Fig. S7). In addition, human PBMCs stimulated with NdlTox, with or without pretreatment with compound A, exhibited no marked inhibition in IL-1β release from NLRC4-activated human PBMCs (Fig. EV2G). These findings collectively demonstrate the selectivity and specificity of compound A in inhibiting the NLRP3 inflammasome over other inflammasomes, while leaving NFκB signaling unaffected.

Next, we compared the potency of compound A to the reference compound MCC950 in BMDMs from C57BL/6N mice. Similar potency was observed in serum-containing conditions in terminally differentiated BMDMs (Fig. 4G). In murine splenocytes stimulated with LPS followed by ATP in the presence of MCC950 or compound A, compound A was found to be at least 6-fold more potent than MCC950 (Fig. 4H). Similarly, in the human PBMC assay, a potency of 27 nM was observed (Fig. 4I). However, in a human whole blood assay, both reference compound MCC950 and

compound A exhibited a significant loss of potency due to high plasma-protein binding (Figs. 4J and EV2H).

Finally, a pharmacokinetic-pharmacodynamic (PK-PD) experiment in wild-type C57BL/6 N was conducted where mice were dosed with 50 mg/kg of compound A and whole blood was collected at 3 and 18 h post dosing. Next, the whole blood was stimulated with LPS + ATP ex vivo and a complete inhibition of IL-1β induction was seen in blood collected 3 h post, indicative of effective NLRP3 inflammasome blockade by circulating compound at that timepoint. Conversely, in blood collected 18 h post dosing, this inhibition of IL-1β was lost owing to the half-life of compound A (Figs. 4K and EV2I,J). In summary, compound A emerged as a selective and potent NLRP3 inhibitor with improved activity compared to reference compound MCC950.

## Compound A shows good in vivo potency and improved efficacy toward CAPS disease

To assess the in vivo efficacy of compound A in inhibiting NLRP3 inflammasome activation, wild-type C57BL/6 N or NLRP3 knock-out mice were orally dosed with 50 mg/kg of reference compound MCC950, compound A, or vehicle PEG400 followed by intraperitoneal injection of 10 mg/ml LPS. After 4 h, vehicle-treated animals exhibited a substantial induction of NLRP3-induced IL-1β levels in the plasma, while NLRP3 knockout animals showed a reduced amount of non-NLRP3-mediated IL-1β (Fig. 5A). Both MCC950 and compound A effectively reduced IL-1β in circulation to the same level as knockout animals, indicating a complete blockade of NLRP3-mediated IL-1β at 50 mg/kg without significant impact on IL-6 and TNF production (Figs. 5A and EV3A,B). Plasma levels of compound A showed higher exposure compared to MCC950 at 4 h post LPS (Fig. EV3C). In a dose–response experiment, oral dosing of 50, 16.7, and 5.6 mg/kg of compound A followed by 10 mg/kg LPS stimulation resulted in a dose-dependent reduction of IL-1β with no marked effect on IL-6 or TNFα (Figs. 5B,C, and EV3D–G). Overall, the in vivo efficacy of compound A was comparable to the reference compound MCC950 in the systemic LPS shock model. In addition, a preclinical rodent 14-day mini-tox experiment using 0, 25, and 100 mg/kg/day revealed no test article-related adverse in-life, clinical pathology, or histopathology findings (organs examined were heart, kidneys, liver, and spleen) in any of the dose groups, indicating that compound A shows no signs of liver or other toxicological markers in this initial preclinical model (Fig. EV3H).

CAPS-associated mutations in *NLRP3* lead to overactivation of the NLRP3 inflammasome resulting in constitutive inflammation

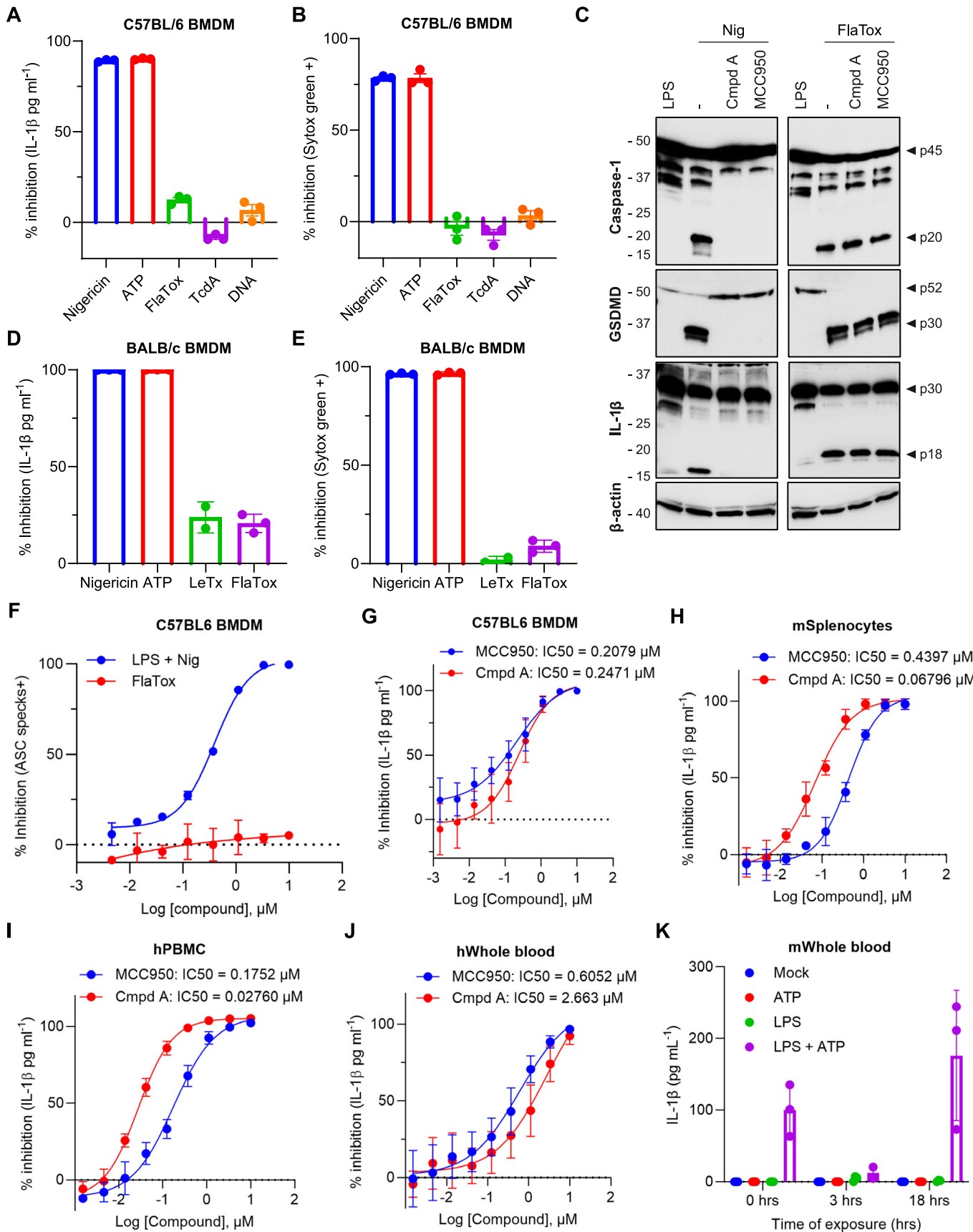

**Figure 4. Full in vitro characterization shows the potent and selective features of compound A.**

(A, B) LPS-primed BMDMs from C57BL/6 mice were treated with compound A (10 μM) followed by stimulation with Nig (1 h), ATP (1 h), FlaTox (2 h), TcdA (5 h), or DNA transfection (5 h). Sytox green-positive cells were counted by incucyte (B), and supernatant collected for IL-1β detection (A). BMDMs from three independent animals were used ($n = 3$) each in triplicate and pooled data shows mean $+/-$ SD. (C) LPS-primed BMDMs from BALB/c mice were treated with compound A or MCC950 (10 μM) followed by stimulation with Nig (1 h) or FlaTox (2 h). Lysates were prepared and ran on western blotting for caspase-1, gasdermin D, IL-1β, and β-actin. BMDMs from three independent animals were used ($n = 3$) and representative blots are shown from matching animal as (D, E). (D, E) LPS-primed BMDMs from BALB/c mice were treated with compound A (10 μM) followed by stimulation with Nig (1 h), ATP (1 h), LeTx (2 h), or FlaTox (2 h). Sytox green-positive cells were counted by incucyte (E), and supernatant collected for IL-1β detection (D). BMDMs from three independent animals were used ($n = 3$) each in triplicate and representative data from 1 animal is shown as mean $+/-$ SD to match the WB. (F) Wild-type BMDMs were treated with LPS + Nig (45 min) or FlaTox (75 min) and ASC specks formation was evaluated in the presence of a dose response of compound A by immunofluorescent staining. BMDMs from two independent animals were used ($n = 2$) each in triplicate and pooled data shows mean $+/-$ SD. (G) LPS-primed BMBMs from C57BL/6 mice were treated with 20 μM Nig in the presence of a dose response of MCC950 or compound A. After 1 h, supernatant was collected and levels of IL-1β were determined using MSD. BMDMs from three independent animals were used ($n = 3$) each in duplicate and pooled data shows mean $+/-$ SD. (H) LPS-primed splenocytes from C57BL/6 mice were treated with 5 mM ATP in the presence of a dose response of MCC950 or compound A. After 1 h, supernatant was collected and levels of IL-1β were determined using MSD. Splenocytes from three independent animals were used ($n = 3$) each in quadruplicate and pooled data shows mean $+/-$ SD. (I) Human healthy donor PBMCs were stimulated with 100 ng/ml LPS for 6 h in the presence of a dose response of MCC950 or compound A. Subsequently, supernatant is used to determine the level of IL-1β using MSD technology. PBMCs from three independent healthy donors were used ($n = 3$) each in quadruplicate and pooled data shows mean $+/-$ SD. (J) Human fresh whole blood was primed with 100 ng/ml LPS for 2 h followed by 5 mM ATP for another 3 h in the presence of compound A. Plasma is collected by centrifugation of the blood for 15 min at 2000 × g and used to determine levels of IL-1β by MSD. Blood from three independent healthy donors was used ($n = 3$) each in quadruplicate and pooled data shows mean $+/-$ SD. (K) C57BL/6 mice were orally dosed with 50 mg/kg compound A and whole blood was collected at indicated timepoints. Plasma is collected after ex vivo stimulation of the blood with 100 ng/ml LPS (2 h) and 5 mM ATP (3 h) and used to determine the level of IL-1β. Per group, three animals were included and data are shown as mean $+/-$ SD at each timepoint. Source data are available online for this figure.

and therefore several symptoms. MCC950 was previously reported to suppress A350V-mediated Muckle–Wells Syndrome (MWS) symptoms (Vande Walle et al, 2019). To evaluate the effect of compound A on CAPS, BMDMs were obtained from wild-type mice and mice carrying the floxed A350V mutation and in vitro exposure to cre-recombination during differentiation was performed. Subsequently, cells were exposed to LPS followed by Nig stimulation either with or without pretreatment with a dose response of compound A (Fig. 5D,E) or MCC950 (Fig. EV4A,B). Compound A showed a 4.5-to-7-fold shift in potency with MWS mutant compared to WT BMDMs. MCC950 showed an even higher shift in potency in the MWS mutant cells, which is in line with previously published observations (Vande Walle et al, 2019; Weber et al, 2022; Molina-López et al, 2024; Cosson et al, 2024). Nonetheless, this suggests that compound A can be used to treat CAPS patients with better potency compared to MCC950 (Figs. 5D,E and EV4A,B).

Based on this shift in potency due to the MWS A350V mutation, an in vivo experiment was conducted using 100 mg/kg. A significant reduction of disease induction was found in animals that received a daily dose of 100 mg/kg compound A as seen by normal bodyweight, decreased IL-1β, IL-18, G-CSF, IP-10, IL-6, and serum amyloid A (SAA) (Figs. 5F–H and EV5A–D). In addition, neutrophilia in MWS-induced mice was reduced after administration of compound A, coinciding with the reduction of identified pathological findings in the animals that received compound A dosing (Figs. 5I and EV5A–F). The livers of vehicle-treated NLRP3 heterozygous A350V knock-in animals exhibited multiple hepatic histopathologic findings that consisted of up to moderate multifocal necrosis/inflammation, thrombosis, infiltration of hepatic sinusoids with mononuclear inflammatory cells, and increased extramedullary hematopoiesis (EMH) (Fig. EV5G). In addition, minimal thrombosis was present in all the lungs of vehicle-treated mice. The treatment of heterozygous mice with 100 mg/kg compound A led to significant reduction in the severity of hepatic necrosis/inflammation, sinusoidal infiltration and thrombosis, minimal reduction in the severity of EMH, and

significant reduction in the incidence of pulmonary thrombosis (Fig. EV5H).

Next, to assess the physiological relevance of CAPS mutations in human settings, THP-1 NLRP3 knockout cells were engineered to express the A354V mutation upon doxycycline treatment. Stimulation of undifferentiated THP-1 A354V cells with either LPS alone or LPS followed by Nig resulted in cell death and release of IL-18 (Figs. 5J,K and EV4C,D). Dose-dependent inhibition on both cell death and IL-18 was seen after treatment with MCC950 and compound A with approximately threefold lower $IC_{50}$ for compound A, indicating a better efficacy of compound A in blocking human overactive NLRP3 (Figs. 5J,K and EV4C,D). To this point, we confirmed the improved IL-1β inhibitory potency of compound A over reference compound MCC950 in primary PBMC from a CAPS-patient diagnosed with Familial Cold Auto-inflammatory Syndrome (FCAS with L355P mutation in human NLRP3) and Neonatal Onset Multisystem Inflammatory Disease (NOMID with D305N mutation in human NLRP3) (Figs. 5L and EV4E). In line within above data, structural overlay shows mutations clustered around the Walker B motif, potentially disrupting ATP hydrolysis (Fig. EV4F). MCC950 may interact with these regions, unlike compound C/A. D305N could affect MCC950 binding through R351, while A354V and L355P in helix351-363 may alter the helix position or stability, impacting MCC950's binding. Compound C/A is likely unaffected by these mutations, as it binds to the opposite side of the pocket (Figs. 2E, EV1D and EV4F). In summary, compound A effectively suppresses inflammasome hyperactivation and associated symptoms related to CAPS, demonstrating notable advantages over reference compound MCC950 in preclinical mouse models as well as under human settings, showcasing its potential as a promising therapeutic candidate.

## Discussion

This study outlines a comprehensive novel screening approach to identify a novel chemical class with high selectivity and potency for

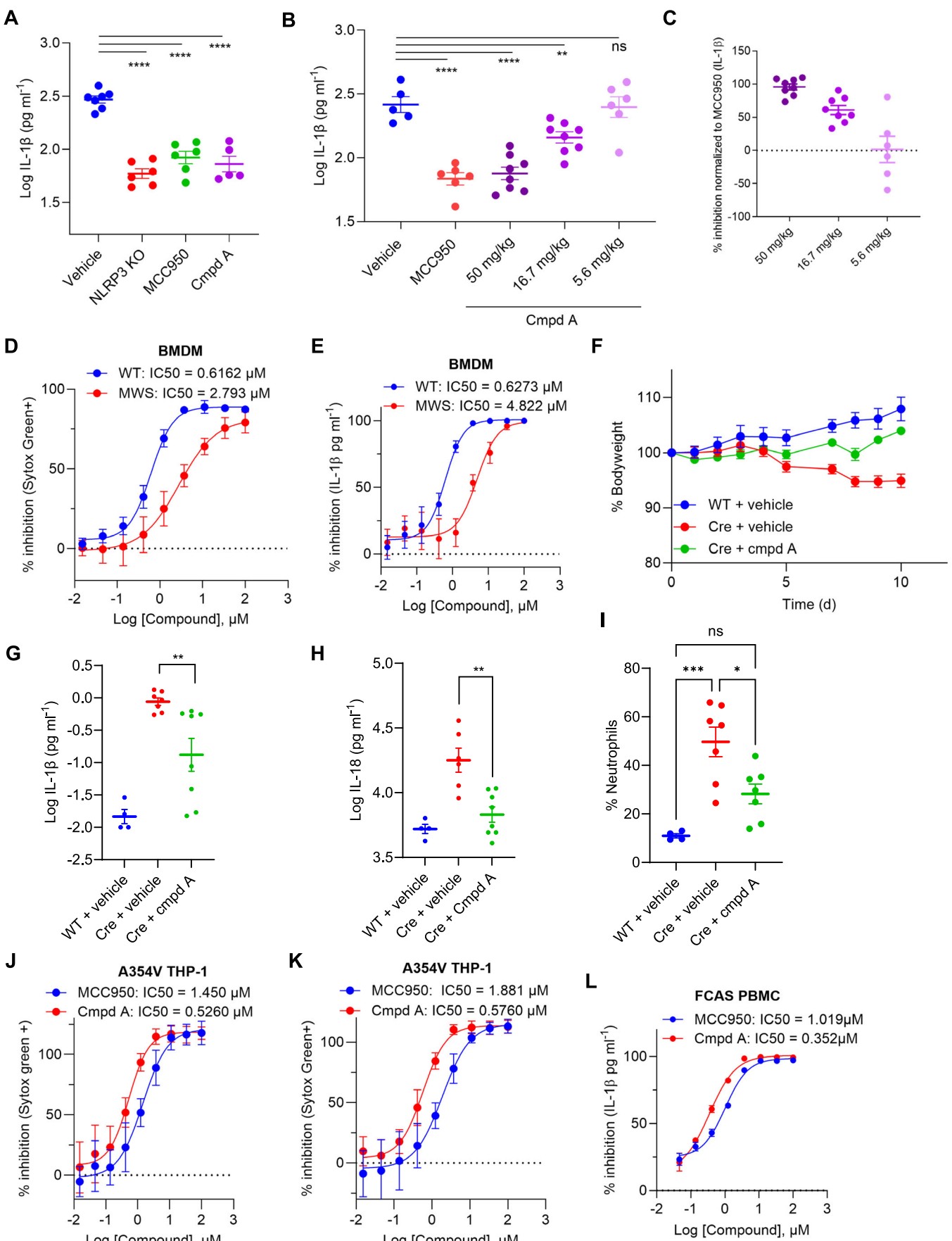

**Figure 5. Full in vivo characterization of compound A shows good potency and improved efficacy toward CAPS disease.**

(A) Wild-type or NLRP3 −/− mice were orally dosed with vehicle, MCC950 or compound A (50 mg/kg) for 30 min followed by an intraperitoneal injection of 10 mg/kg LPS. After 4 h, the mice are euthanized, and the blood is collected for IL-1β cytokine determination. Per group, 8 animals are used and mean +/− SEM is depicted. One-way ANOVA with Dunnett's multiple comparison test was performed: ****$P < 0.0001$ in all groups. (B) Wild-type mice were orally dosed with vehicle, MCC950 (50 mg/kg) or compound A (50–16.7–5.6 mg/kg) for 30 min followed by an intraperitoneal injection of 10 mg/kg LPS. After 4 h, the mice are euthanized, and the blood is collected for IL-1β cytokine determination. Per group, 8 animals are used and mean +/− SEM is depicted. One-way ANOVA with Dunnett's multiple comparison test was performed: MCC950 group: ****$P < 0.0001$, 50 mg/kg group: ****$P < 0.0001$, 16.7 mg/kg group: **$P = 0,0039$, 5.6 mg/kg group: ns. (C) % inhibition of IL-1β from (B) calculated by normalization to MCC950. Per group, eight animals are used and mean +/− SEM is depicted. (D) BMDMs from wild-type or A350V +/- MWS mice treated with TAT-cre were stimulated with LPS + Nig in the presence of a dose response of compound A and Sytox green (1 μM) positive cells were determined by Incucyte. BMDMs from three independent animals were used ($n = 3$) each in duplicate and pooled data shows mean +/− SD. (E) BMDMs from wild-type or A350V +/− MWS mice treated with TAT-cre were stimulated with LPS + Nig in the presence of a dose response of compound A and supernatant was used to determine the level of IL-1β using MSD. BMDMs from three independent animals were used ($n = 3$) each in duplicate and pooled data shows mean +/− SD. (F) Wild-type or A350V +/− mice were orally treated with tamoxifen for 5 d combined with a daily dose of compound A (100 mg/kg). Bodyweight was determined daily. Per group, four (WT vehicle), seven (Cre + vehicle), or eight (Cre + Cmpd A) animals were used and mean +/− SEM is depicted. (G) Wild-type or A350V +/− mice were orally treated with tamoxifen for 5 d combined with a daily dose of compound A (100 mg/kg). On day 10, animals were euthanized, and plasma was used to determine levels of IL-1β using high-sensitivity Quanterix. Per group 4 (WT vehicle), 7 (Cre + vehicle) or 8 (Cre + Cmpd A) animals were used and mean +/− SEM is depicted. One-way ANOVA with Bonferroni's multiple comparison test was performed: Cre cmpd A group: **$P = 0.0013$. (H) Wild-type or A350V +/− mice were orally treated with tamoxifen for 5 d combined with a daily dose of compound A (100 mg/kg). At day 10, animals were euthanized, and plasma was used to determine levels of IL-18 using Luminex. Per group 4 (WT vehicle), 7 (Cre + vehicle) or 8 (Cre + Cmpd A) animals were used and mean +/− SEM is depicted. One-way ANOVA with Bonferroni's multiple comparison test was performed: Cre cmpd A group: **$P = 0.0011$. (I) Wild-type or A350V +/− mice were orally treated with tamoxifen for 5 d combined with a daily dose of compound A (100 mg/kg). At day 10, animals were euthanized, and blood was used to determine the number of neutrophils by Sysmex. Per group, four (WT vehicle), seven (Cre + vehicle), or eight (Cre + Cmpd A) animals were used, and mean +/− SEM is depicted. One-way ANOVA with Bonferroni's multiple comparison test was performed: Cre vehicle group: ***$P = 0.0005$, Cre cmpd A group: *$P = 0.0159$. (J) Doxycyclin-inducible A354V THP-1 cells were treated with LPS in the presence of a dose response of MCC950 or compound A. Sytox green-positive cells were determined using Incucyte. Two independent repeats ($n = 2$) were performed in quadruplicate and pooled data depicted as mean +/− SD. (K) Doxycyclin-inducible A354V THP-1 cells were treated with LPS + Nig in the presence of a dose response of MCC950 or compound A. Sytox green-positive cells were determined using Incucyte. Two independent repeats were performed in quadruplicate and pooled data depicted as mean +/− SD. (L) PBMCs from a patient with confirmed FCAS diagnosis were pretreated with a dose response of MCC950 or compound A followed by exposure to LPS + Nig for 6 h. Supernatant was used to determine the level of IL-1β by MSD. PBMCs were treated in duplicated, and data represented as mean +/− SD. Source data are available online for this figure.

NLRP3, particularly focusing on the direct binding to the NACHT domain of human NLRP3. To avoid potential off-target effects on NFκB observed by competitors (Zhang et al, 2023), we implemented a primary screening assay using a murine macrophage cell line (J774A.1) with a two-step activation system of the NLRP3 inflammasome. In addition, counter-screening against NLRC4 was used to eliminate common downstream pathway inhibitors. Given the limited sequence homology between NLRP3 and NLRC4, it is unlikely that inhibitors would directly target both inflammasomes. However the downstream pathways are highly conserved, so any compound able to inhibit in both assays is more than likely an off-target effect downstream of NLRP3 and undesired (Paerewijck and Lamkanfi, 2022).

Simultaneously, in collaboration with the Target Discovery Institute at Oxford (UK), a phenotypic high-content ASC speck imaging screening assay was established. This screening employed the 81 K compound JumpStarter library to identify novel NLRP3 inhibitors. Markedly, the inclusion of several annotated compounds with known target provided valuable insights into NLRP3 pathway biology (Nizami et al, 2021). Unfortunately, they did not identify inhibitors on-target, but those inhibiting the pathway, owing to their screening method based on ASC specks rather than the extensive triage funnel. Notably, using this funnel with additional inflammasome assays, in-house promiscuity data, and target engagement, we identified only 1 cluster with the desired MOA from a screen of almost 1 million compounds.

Following our initial screening in the murine cell line, the study progressed to human PBMCs to assess potency in a human cellular system and achieve selectivity across a broad inflammasome panel. We identified cluster 1, which exhibited a preference for inhibiting

NLRP3 with a good potency in the human PBMC assay. During the preparation of this manuscript, three papers were published describing the tricyclic compounds identified as hits in this screening campaign confirming the robust design of the screening funnel (Vande Walle et al, 2024; Velcicky et al, 2024; Li et al, 2023).

The reference compound MCC950 is known to block the ATPase activity of NLRP3 by binding in close proximity to the Walker A, and B motifs (Vande Walle et al, 2019; Coll et al, 2019; Tapia-Abellán et al, 2019). Building on this knowledge, we investigated the effect of cluster 1 on the ATPase activity and observed a dose-dependent inhibition, suggesting a common MOA to MCC950. After a significant triage, it was clear that a cellular phenotypic screen with appropriately designed counter screens to remove those compounds acting on the complex pathway of NLRP3 could identify protein-selective inhibitors for NLRP3.

To further validate target engagement, we employed Cryo-EM, nanoDSF and HDX techniques. These analyses confirmed the interaction of cluster 1 compounds with the recombinant MBP-hNLRP3-ΔPYD, providing evidence of target engagement. Several small molecules have been shown to effectively inhibit NLRP3 activation, with seven having their structures characterized (Dekker et al, 2021; Hochheiser et al, 2022; Ohto et al, 2022; McBride et al, 2022; Velcicky et al, 2024; Yu et al, 2024; Ohba et al, 2023). These molecules bind a deeply embedded pocket in the NACHT domain, located near the ATP/ADP site and the Walker A motif, and interact with all NACHT subdomains thereby confining NLRP3 to the closed conformation state (Fig. EV1A–C). They share a common structural feature: a central amide moiety with a bulky head and small charged tails extending toward the LRR region

(Fig. EV1B). The bulky heads overlap within a well-defined pocket, while the central amide moiety interacts either with Arg351 (as seen in compounds 8GI, MCC950, WTN, RM5, and XE3) or with Arg578 (in compound C and A1H02) (Fig. EV1C,D).

The interaction of compound C with NLRP3 differs from that of MCC950, primarily due to the amide moiety occupying a different sub-pocket, replacing the undesirable anionic sulfonylurea present in MCC950. Compounds C and A1H02 share notable similarities, such as a tricyclic head, central amide moiety, and a single-ring tail, with similar binding modes to NLRP3-NACHT (Fig. EV1B,D). Although compound C has a smaller accessible surface area, its comparable binding region to NLRP3-NACHT highlights its strong potential as a reference for advancing this compound series (Fig. EV1B,E). These structural insights may contribute to a better understanding of the compound's specificity and its potential for therapeutic intervention in the NLRP3 pathway.

Using cluster 1 as a starting point, a chemistry-driven SAR campaign was designed to improve the physical-chemical properties of the molecule enabling in vivo PoC. This was successfully achieved with compound E demonstrating comparable efficiency to the reference compound. However, the tricycle core suffered from poor metabolic stability and solubility leading to the fast evolution towards a bicycle phthalazine to enable improved metabolic stability and solubility (Velcicky et al, 2024; Li et al, 2023). Further optimization resulted in the identification of our lead molecule compound A with good potency, decent solubility, clearance, and permeability. Compound A was extensively characterized both in vitro, in human and murine cell systems, and in in vivo murine animal models. When comparing compound A to the reference compound MCC950, we found improved potency in human PBMCs (27 nM vs 175 nM, Fig. 4I), murine splenocytes (68 nM vs 440 nM, Fig. 4H) and demonstrated augmented activity towards cells expressing NLRP3 containing CAPS mutations regardless of the single nucleotide polymorphism (SNP) present in human or mouse gene (Figs. 5D,E,J–L and EV4). In addition, in a CAPS in vivo model, we were able to prevent disease symptoms with a daily dose of compound A. Remarkably, in our preclinical rodent toxicity model, the phthalazine compound A demonstrated an absence of adverse signs related to hepatotoxicity over a 14-day observation period. However, a comprehensive toxicological assessment directly comparing the two chemical classes would be required to verify these findings in relation to the diaryl sulfonylurea-related compounds, where previous investigations have reported hepatotoxic effects (Charan et al, 2023; Shah et al, 2015; Mangan et al, 2018).

In conclusion, this screening campaign successfully led to the identification and characterization of a novel chemical class of NLRP3 inhibitors that exhibit high selectivity and potency. The compounds derived from this class have shown remarkable efficacy in addressing CAPS, marking a significant advancement in the field. The findings support the feasibility of clinical development for this series of compounds, with a proposed initial focus on CAPS patients to establish a clear proof-of-concept for NLRP3-mediated therapeutic effects. Further clinical trials and extended studies are warranted to fully elucidate the safety, efficacy, and potential broader applications (eg. cancer, neuroinflammation) of these promising NLRP3 inhibitors, laying the foundation for their translation into novel therapeutic interventions.

## Methods

### Reagents and tools table

| Reagent/resource | Reference or source | Identifier or catalog number |
|---|---|---|
| **Experimental models** | | |
| C57BL/6N (*M. musculus*) | Charles River Laboratories | #027 |
| BALB/c (*M. musculus*) | Charles River Laboratories | #028 |
| floxed NLRP3 (*M. musculus*) | J&J | N/A |
| Deleter Cre (*M. musculus*) | Taconic | #12524 |
| NLRP3$^{A350VneoR}$ (*M. musculus*) | Jax Laboratories | #17969 |
| Tamoxifen-inducible CreERT2 (*M. musculus*) | J&J | N/A |
| J774A.1 cells (*M. musculus*) | ATCC | TIB-67 |
| L929 cells (*H. sapiens*) | ATCC | CCL-1 |
| THP-1 cells (*H. sapiens*) | Invivogen | thp-nullz |
| NLRP3 knockout THP-1 cells (*H. sapiens*) | Invivogen | thp-konlrp3z |
| PBMC (*H. sapiens*) | In-house sourced | N/A |
| Sf9 cells (insect) | Expression Systems | 94-001F |
| **Recombinant DNA** | | |
| pLV[Exp]-Puro-EF1A > {Asc-3x(SGGGG-HA)-mcherry} | VectorBuilder | N/A |
| pLV[Exp]-Puro-TRE3G > {hNLRP3-6Ala-3xFLAG MWS A354V} | VectorBuilder | N/A |
| PVL1393 vector | Expression Systems | 91-013 |
| **Antibodies** | | |
| Mouse anti-mouse caspase-1 | Adipogen | AG-20B-0042-C100 (Casper-1) |
| Rabbit anti-mouse Gsdmd (full lenght) | Abcam | #219800 (EPR20859) |
| Rabbit anti-mouse Gsdmd (cleaved) | Abcam | #209845 (EPR19828) |
| Rabbit anti-mouse IL-1β | Genetex | GTX74034 |
| Mouse anti-mouse NLRP3 | Adipogen | AG-20B-0014-C100 (Cryo-2) |
| Rabbit anti-mouse ASC | Adipogen | AG-25B-0006-C100 (pAL177) |
| Mouse anti-β-actin | Santa Cruz | sc-47778 HRP (C4) |

| Reagent/resource | Reference or source | Identifier or catalog number |
|---|---|---|
| Goat anti-Mouse IgG (H + L) Secondary Antibody, HRP | Thermo Fisher | 31430 |
| Goat anti-Rabbit IgG (H + L) Secondary Antibody, HRP | Thermo Fisher | 31460 |
| Goat anti-Rabbit IgG (H + L) Secondary Antibody | Thermo Fisher | A-11034 |
| Hoechst | Invitrogen | H3570 |
| CellMask Deep Red | Thermo Fisher | H32721 |
| **Chemicals, enzymes, and other reagents** | | |
| DMEM | Sigma | D5796-6x500ML |
| IMDM | Thermo Fisher | 12440053 |
| RPMI | Sigma | R0883-6x500ML |
| DPBS | Thermo Fisher | 14040133 |
| OptiMEM | Life Technologies | 31985-047 |
| HEPES | Sigma | H0887-100ML |
| NaCl (5 M) | Sigma | S6546-1L |
| MgCl2 (1 M) | Sigma | M1028-100ML |
| CHAPS | Calbiochem | 3055-100GM |
| TCEP (0.5 M) | Thermo Scientific | 77720 |
| Glycerol | Sigma | G9012-1L |
| Fetal bovine serum | Tico Europe | #6Q4102388065-000010# |
| Pen/Strep | Life Technologies | 15070-063 |
| CellStripper | Corning | 25-056-CI |
| MEM EAGLE NON ESSENTIAL - 100 ML | VWR | LONZ13-114E |
| L-Glutamine | Sigma | G7513-100ML |
| *E. coli* Lipopolysaccharide serotype O111:B4 | Sigma | L4130 |
| Nigericin | Sigma | N7143 |
| ATP | Roche | 10519987001 |
| B. anthracis protective antigen (PA) | PepCore VIB Gent | N/A |
| B. anthracis lethal factor (LF) | Quadratech | 172 C |
| flagellin of *L. pneumophilia* (LFn-FlaA) | PepCore VIB Gent | N/A |
| Pam3CSK4 | Invivogen | tlrl-pms |
| LFn fused to *B. thailandensis* T3SS Needle (LFn-Ndl) | Invivogen | tlrl-ndl |
| *C. difficile* toxin A (TcdA) | Enzo Life Science | ENZ-PRT271-0002 |
| lipofectamine 2000 | Invitrogen | 11668019 |

| Reagent/resource | Reference or source | Identifier or catalog number |
|---|---|---|
| Doxycycline | Sigma | D9891 |
| Complete EDTA protease inhibitors | Roche | 4693159001 |
| Sytox green | Invitrogen | S7020 |
| AlphaLISA Mouse IL-1β Detection Kit | Perkin Elmer (Revvity) | AL503C |
| Kit for IL-1β, IL-6 and TNF detection | MesoScale Discovery | K151A9H |
| Human IL-18 detection | Procartaplex, Thermo Fisher | EPX01A-10267-901 |
| Human IL-1β detection | Procartaplex, Thermo Fisher | EPX01A-10224-901 |
| Human IL-6 detection | Procartaplex, Thermo Fisher | EPX01A-10213-901 |
| Human TNF detection | Procartaplex, Thermo Fisher | EPX01A-10223-901 |
| Murine IL-18 detection | Procartaplex, Thermo Fisher | EPX01A-20618-901 |
| Murine IL-1β detection | Procartaplex, Thermo Fisher | EPX01A-26002-901 |
| Murine IL-6 detection | Procartaplex, Thermo Fisher | EPX01A-20603-901 |
| Murine TNF detection | Procartaplex, Thermo Fisher | EPX01A-20607-901 |
| Murine G-CSF detection | Procartaplex, Thermo Fisher | EPX01A-26034-901 |
| Murine IP-10 detection | Procartaplex, Thermo Fisher | EPX01A-26018-901 |
| Murine IL-1β ELISA | R&D | SMLB00C |
| 4x Laemmli Sample Buffer | Bio-rad | #1610747 |
| tamoxifen | Sigma | T5648 |
| Murine IL-1β | Quanterix | #102517 |
| Murine Serum Amyloid A ELISA | Abcam | ab215090 |
| SuperSignal West Pico PLUS | Thermo Fisher | 34578 |
| MagicMark XP Western Protein Standard | Thermo Fisher | LC5602 |
| SeeBleu Plus2 Pre-Stained Protein Standard | Thermo Fisher | LC5925 |
| amylose resin | New England BioLabs Inc. | E8021L |
| Superose 6 Increase 10/300 GL | Cytiva | 29-0915-96 |
| Quantifoil® R 1.2/1.3 300 Mesh, Au | Quantifoil | Q3100AR1.3 |
| C-Clip Ring (100x) | Thermo Fisher | 1036173 |
| C-Clip (100x) | Thermo Fisher | 1036171 |
| Disuccinimidyl suberate | Sigma | S1885 |
| **Software** | | |

| Reagent/resource | Reference or source | Identifier or catalog number |
|---|---|---|
| HDExaminer 3.3 | Sierra Analytics | |
| IncuCyte S3 software | Sartorius | |
| Graphpad Prism 10 | | |
| Relion 4 | | |
| Coot 0.95 | | |
| Phenix 1.18.2 | | |
| Pymol 2.5.0 | | |
| Schrodinger 2024-3 | | |
| CCP4 7.1 | | |
| UCSF Chimera 1.18 | | |
| Signals Image Artist | | |
| **Other (equipment...)** | | |
| Dionex RSLC | ThermoFisher | |
| LTQ Velos Orbitrap MS | ThermoFisher | |
| protease type XVIII/ pepsin column | NovaBioassays | |
| Mascot v 2.6 | ThermoFisher | |
| PheraSTAR FSX | BMG | |
| Glacios Microscope | ThermoFisher | |
| Titan Krios Microscope | ThermoFisher | |
| PELCO easiGlow | TED PELLA, INC | |
| Vitrobot Mark IV | ThermoFisher | |
| CellVoyager CV8000 | Yokogawa | |
| **DNA material** | | |
| CreERT2 RoSQ 6 Fw | IDT | CTGTTGGGCACTGACAATTCCGTG |
| CreERT2 RoSQ 2 Rev | IDT | TGCTTACATAGTCTA ACTCGCGAC |
| Nlrp3 Jacks KI WT Fw/Fcas+MWS | IDT | CACCCTGCATTTTGT TGTTG |
| Nlrp3 Jacks KI Mut Fw/Fcas+MWS | IDT | GCTACTTCCATTTGT CACGTCC |
| Nlrp3 Jacks KI rev/ Fcas+MWS | IDT | CGTGTAGCGACTGTTGAGGT |
| Cond Nlrp3 KO FW | IDT | ACACCAGAATTTTGGGAGCCT |
| Cond Nlrp3 KO Rev | IDT | TGGTATGACCGGACAGAGGG |
| Nlrp3 KO 2 Rev | IDT | CCCTAGCTTTCAAAAAGAGTTGA |
| Cre 1 Rev | IDT | GGAAAATGCTTCTGT CCGTTTGC |
| Cre 1 FW | IDT | ATTGCTGTCACTTGG TCGTGGC |

## Mice

C57BL/6N were purchased from Charles River Laboratories and housed in individually ventilated cages under specific pathogen-free conditions. All studies were conducted under protocols approved by the local Ethical Committee of Johnson & Johnson on the Use and Care of Animals. Mice expressing conditional targeted *Nlrp3* were obtained by homologous recombination introducing the targeting cassette around exon 5. The floxed NLRP3 mice were crossed with Deleter Cre mice (Taconic, #12524) thereby creating whole-body NLRP3 knockout animals that were validated by genotyping and on in vitro functionality (Appendix Fig. S8). NLRP3$^{A350VneoR}$ mice were purchased from Jax Laboratories (#017969). Tamoxifen-inducible CreERT2-expression was introduced under the control of the ROSA26 promotor in C57BL/6N mice. Tamoxifen-induced recombination was validated by genotyping (Appendix Fig. S9).

## Cells

J774A.1 cells (ATCC) were maintained in T225 tissue culture flasks using DMEM containing 25 mM HEPES supplemented with 10% fetal bovine serum (FBS), 100 IU/mL penicillin and 100 μg/mL streptomycin and harvested using CellStripper. Bone marrow-derived macrophages (BMDMs) were generated from both male and female mice aged between 7 and 20 weeks. Briefly, bone marrow cells were cultured in L929-cell-conditioned IMDM supplemented with 10% FBS, 1% nonessential amino acids and 1% penicillin–streptomycin for 6 days in a humidified atmosphere containing 5% $CO_2$. For experiments, BMDMs were detached, seeded in 96 wells at $5 \times 10^5$ cells per ml and allowed to attach overnight in a 5% $CO_2$ incubator at 37 °C. Murine splenocytes were obtained by mechanical disruption of the spleen and were cultured in DMEM supplemented with 10% FBS, 1% glutamine and 1% penicillin–streptomycin and used immediately at $8.5 \times 10^6$ cells per ml in a 96-well and kept at 5% $CO_2$ and 37 °C.

Fresh human blood was collected from healthy individuals in lithium-heparin tubes and used at a 1:1 ratio with RPMI in a 96-well plate at 5% $CO_2$ and 37 °C. For murine blood, animals were terminally sedated, and blood was collected via cardiac puncture in lithium-heparin tubes and used undiluted in 96-well format at 5% $CO_2$ and 37 °C. Isolation of PBMC from healthy blood donor buffy coat using Ficoll–Hypaque density gradient centrifugation was performed and PBMCs were stored in liquid nitrogen until further use. Upon thawing, PBMCs were resuspended in culture medium consisting of RPMI supplemented with 10% FBS, 1% glutamine, and 1% penicillin–streptomycin. After cell viability was determined, cells were seeded at a density of $1 \times 10^6$ cells per ml in 96-well plate and were maintained in a 5% $CO_2$ incubator at 37 °C. THP-1-ASC-mCherry were created by transducing wild-type THP-1 cells (Invivogen, thp-nullz) with a lentivirus to randomly introduce ASC-mCherry in the genome. Bulk population was sorted to select mCherry-positive cells and functional validation was performed. NLRP3 knockout THP-1 cells (Invivogen, thp-konlrp3z) were transduced with a lentivirus introducing NLRP3 containing the point mutation A354V under control of a doxycycline promoter under puromycin selection. NLRP3 KI was confirmed by Sanger sequencing (Top: WT, bottom: MWS KI) (Appendix Fig. S10). Human PBMCs from FCAS and NOMID patients were obtained via Sanguine Bio.

## Stimulations

Cells were either left untreated or primed with 100 ng /ml E. coli LPS (Sigma, L4130) for 2 or 5 h followed by inflammasome-specific

triggers. For the murine NLRP3 inflammasome, LPS-primed cells were treated with 20 μM Nig (Sigma, N7143) or 5 mM ATP (Roche, 10519987001) for 1 h. In human PBMCs, NLRP3 inflammasome was stimulated using 100 ng/ml LPS for 6 h. To activate the NLRP1b inflammasome, LPS-primed cells were treated with the combination (LeTx) of *B. anthracis* protective antigen (PA, 2 μg/ml, PepCore VIB Gent) and *B. anthracis* lethal factor (LF, 1 μg/ml, Quadratech, 172C). To stimulate the NLRC4 inflammasome, murine LPS-primed cells were treated with the combination (FlaTox) of anthrax protective antigen (PA, 2 μg/ml, PepCore VIB Gent) and the fusion protein of the N-terminal piece of anthrax lethal factor with flagellin of *L. pneumophilia* (LFn-FlaA, 2 μg/ml, PepCore VIB Gent). In human cells, a 3 h-priming with 1 μg/ml Pam3CSK4 (Invivogen, tlrl-pms) was followed by treatment with the combination (NdlTox) of PA and LFn fused to *B. thailandensis* T3SS Needle (LFn-Ndl, Invivogen, tlrl-ndl) for another 3 h. For triggering the PYRIN inflammasome, LPS-priming was followed by 200 μg/ml C. difficile toxin A (TcdA, Enzo Life Science) for 5 h and for the AIM2 inflammasome, LPS-primed cells were transfected with dsDNA using lipofectamine 2000 (Invitrogen, 11668019) for 5 h according to manufacturer's protocol. Undifferentiated THP-1 MWS cells were activated with 1 ng/ml doxycycline (Sigma, D9891) and the next day treated with 100 ng/ml LPS or LPS + Nig for indicated time points. Doxycycline concentration was optimized to prevent autoactivation of MWS KI.

## Recombinant proteins

All purifications were done at 4 °C. DNA encoding NLRP3_130_1036 (NLRP3 ΔPYD, UniProtQ96P20) was engineered with MBP-TEV at the N-terminus and inserted into a PVL1393 vector (Epoch Life Sciences). Virus was generated and used to infect Sf9 insect cells. MBP-NLRP3-ΔPYD was released from cell paste by sonication in 25 mM HEPES pH = 7.4 0.4 M NaCl 0.4% CHAPS 20% glycerol 0.2 mM TCEP with Complete EDTA protease inhibitors. MBP-NLRP3-ΔPYD construct was captured from clarified lysate supernatant by amylose resin (New England BioLabs Inc). The amylose resin was washed with 2 M NaCl followed by 10 mM ATP-MgCl₂ in lysis buffer and eluted with 20 mM maltose. MBP-NLRP3-ΔPYD was further purified by Superose 6, the retained peak was pooled for target engagement assays.

For Cryo-EM, NLRP3.136_1036.R137C.K138A.K142A was designed, expressed, and purified as described by the lab of Hao Wu (Sharif et al, 2019) with minor modifications. A flag.6His tag was engineered onto the N-terminus of the MBP, and 50 μM compound +2 mM ADP were incubated overnight prior to the Superose 6 run into 25 mM HEPES pH = 7.4 0.4 M NaCl 0.2% CHAPS 0.2 mM TCEP.

## HDX-MS

HDX-MS experiments were carried out as described previously (Diaz et al, 2022) with minor modifications. In summary, NLRP3 was sequenced using 12 μM undeuterated protein, diluted fourfold in protein storage buffer and quenched in fourfold excess of 2 M Urea, 0.8% formic acid and 20 mM TCEP (quench buffer). Ligand-bound states were prepared by mixing saturating concentrations of the ligand and incubating on ice for 15 min. HDX-MS workflow for deuterated samples differed in the composition of dilution buffer, which was prepared in D₂O. Reactions were stopped at three time points (10, 100, and 1000 s) in ice-cold Buffer Q and flash-frozen.

LC-MS was performed using Dionex RSLC (ThermoFisher) interfaced with LTQ Velos Orbitrap MS (ThermoFisher). Once thawed, the samples were pushed through protease type XVIII/pepsin column (NovaBioassays) and onto a trap column (self-packed Poros R10, 2.1 × 4 cm). Digested peptides were separated on a BioZen 2.6 μM peptide XB-C18 column (50 × 2.1 mm) at 5 °C. The mass was measured with resolution of 60,000 and mass range of 300-2000 with top5 MS/MS in ion trap with dynamic exclusion for 30 s. Sequencing files for undeuterated protein were generated using Proteome Discoverer 2.1 and searched using Mascot v 2.6 (ThermoFisher). N-terminal acetylation was added as fixed modification and the enzyme was set as non-specific. HDX data analysis was performed using HDExaminer 3.3 (Sierra Analytics) with manual inspection.

## Grid preparation and data acquisition

In total, 3.5 μL of 4.4–4.6 mg/ml purified NLRP3 compound C complex was applied to the plasma-cleaned (Gatan Solarus) Quantifoil 1.2/1.3 holey gold grid, and subsequently vitrified using a Vitrobot Mark IV (FEI Company). Grids were loaded into a Titan Krios transmission electron microscope (ThermoFisher Scientific) with a post-column Gatan Image Filter (GIF) operating in nanoprobe at 300 keV with a Gatan K2 Summit direct electron detector and an energy filter slit width of 20 eV. Images were recorded with Leginon in counting mode with a pixel size of 1.04 Å and a nominal defocus range of −0.8 to −1.5 μm. Images were recorded with a 6 s exposure and 200 ms subframes (30 total frames) corresponding to a total dose of ~44.57 electrons per Å². All details corresponding to individual datasets are summarized in (Appendix Fig. S3).

## Electron microscopy data processing

Dose-fractioned movies were gain-corrected, and beam-induced motion correction using MotionCor2 with the dose-weighting option. The particles were automatically picked from the dose-weighted, motion-corrected average images using Relion 3.0 (Zivanov et al, 2018). CTF parameters were determined by Gctf. Particles were then extracted using Relion 3.0 with a box size of 300 pixels. The 3D classification and refinement were performed with Relion 3.0 using the binned datasets. One round of 3D classification was performed to select the homogenous particles. Unbinned homogenous particles were re-extracted and then submitted to 3D auto-refinement with D2 symmetry imposed. 3D classifications and 3D refinements were started from a 60 Å low-pass filtered version of an ab initio map generated with Relion 3.0. All resolutions were estimated by applying a soft mask around the protein complex density and based on the gold-standard (two halves of data refined independently) FSC = 0.143 criterion. Prior to visualization, all density maps were sharpened by applying different negative temperature factors using automated procedures, along with the half-maps, were used for model building. Local resolution was determined using ResMap (Appendix Fig. S4).

## Model building and refinement

The initial template of the NLRP3 was derived from a homology-based model calculated by SWISS-MODEL. The model was docked

into the EM density map using Chimera (Pettersen et al, 2004) and followed by manual adjustment using COOT (Emsley et al, 2010). Note that the EM density around the MBP regions was poor relative to other parts of the model. Each model was independently subjected to global refinement and minimization in real space using the module phenix.real_space_refine in PHENIX (Afonine et al, 2018) against separate EM half-maps with default parameters. The model was refined into a working half-map, and improvement of the model was monitored using the free half-map. Model geometry was further improved using Rosetta (Wang et al, 2016). The geometry parameters of the final models were validated in COOT and using MolProbity and EMRinger. These refinements were performed iteratively until no further improvements were observed. The final refinement statistics were provided in (Appendix Fig. S3). Model overfitting was evaluated through its refinement against one cryo-EM half-map. FSC curves were calculated between the resulting model and the working half-map as well as between the resulting model and the free half and full maps for cross-validation (Appendix Fig. S4). Figures were produced using PyMOL and Chimera.

## Nanoscale differential scanning fluorometry (NanoDSF)

Prometheus NT.48 (NanoTemper Technologies GmbH, Munich, Germany) was used to analyze protein thermal stability in response to ligand binding. Purified recombinant human NLRP3 (1 µM) was incubated at room temperature with 100 µM of indicated compound, ±100 µM ADP (adenosine 5'-diphosphate, Sigma 01905) in 12 µL reaction volume for 30 min in reaction buffer containing 25 mM HEPES pH 7.4, 400 mM NaCl, 0.2% CHAPS, 0.2 mM TCEP, the final % DMSO in the well is ≥1% after compound addition. Assays were prepared in Greiner 384 well non-binding black plates (Greiner 784900) and transferred to the instrument using the Prometheus NT.Plex nanoDSF Grade Standard Capillary Chips (PR-AC002). The instrument excites samples at 280 nm and records intrinsic protein fluorescence at 330 and 350 nm, the lambda maxes associated with buried and exposed tryptophan residues, respectively. Measurements were taken over a 20–95 °C thermal gradient with a 1 °C per minute ramp rate. The 350 nm/330 nm fluorescence ratio was plotted versus temperature and the first derivative of this plot was used to determine the melting temperature (TM) under each condition. Data analysis was done by PR.ThermControl, version 2.1.6 by NanoTemper.

## NLRP3 ATPase activity and inhibition

For ATPase activity assay, purified recombinant human NLRP3 (0.5 µM) was incubated at room temperature with indicated concentration of MCC950 for 30 min in the reaction buffer containing 25 mM HEPES pH 7.4, 0.2 mM NaCl, 5 mM MgCl$_2$, 0.01% CHAPS, 0.1 mM TCEP. ATP (100 µM, Ultra-Pure ATP) was then added, and the mixture was further incubated at 37 °C for another 5 h. The amount of ATP converted into adenosine diphosphate (ADP) was determined by luminescent ADP detection with ADP-Glo Kinase Assay kit (Promega, Madison, MI, USA) according to the manufacturer's protocol. The results were expressed as percentage of residual enzyme activity to the vehicle-treated enzyme.

## Molecular dynamic simulation

Explicit solvent MD simulations were conducted on NLRP3, both in the presence and absence of the compound, to examine its flexibility and dynamics in solution. The starting monomeric NLRP3 structure was derived from the Cryo-EM complex with compound C. The initial models for compounds C and A were generated from their SMILES files, processed using LigPrep in Schrodinger 2021-1 (www.schrodinger.com), and then aligned to compound C within its binding pocket in the NACHT domain of the cryo-EM complex structure.

Three separate simulations were carried out: one for NLRP3 alone (referred to as "Apo"), one with compound C, and another with compound A. These simulations excluded the effects of oligomerization and the presence of ATP or ADP. To enhance sampling REST (replica exchange with solute tempering) MD (Liu et al, 2005) simulations were performed. Eight replicas were used and the simulations were run at 300 K and 1 bar. Each replica was run for 10 ns using the Desmond simulation package (KJ, 2006) in Schrodinger 2021-1 (www.schrodinger.com). The systems were protonated at neutral pH and centered in a cubic box such that the minimum distance from any protein atom to the box wall was 10 Å. The box was solvated using simple point-charge (SPC) (Robinson et al, 1996) (water molecules and counter ions were added to neutralize the system. OPLS4 force field (Lu et al, 2021) was used as the potential energy function for the protein. The default relaxation protocol in Maestro was employed prior to the production simulations.

Three simulation frames (Frame 50, 500, 1000) were selected and overlaid with the compound C-bound NLRP3 structure. Despite some variations, key interactions between the compounds and NLRP3 residues (WHD-Thr439, HD2-Phe575, Arg578, Glu629, Tyr632, Glu636, LRR-Met661, and Asp662) remained consistent (Fig. 3F). The variations observed between the simulations and the compound C-bound cryo-EM structure could stem from several factors: (1) ATP/ADP were excluded during the simulations, (2) oligomerization states were not considered since monomers were used to reduce computational load, and (3) the simulations may have captured intermediate states not observable by cryo-EM or X-ray methods.

## Cell death analysis

IncuCyte S3 (Sartorius) was used to quantify cell permeabilization in the presence of 5 µM Sytox green (Invitrogen, S7020).

## ASC speck assay

BMDMs were fixed using 4% formaldehyde and permeabilized using 0.1% Triton-X in PBS solution for 30 min. Next, 1% BSA blocking buffer was incubated for 1 h, followed by staining with anti-ASC antibody in blocking buffer (1/200, pAL177, Adipogen) for 1 hr. Goat anti-rabbit IgG (H + L) (1/500, A-11034, Thermo-Fisher) secondary antibody was combined with Hoechst (1/1000, H3570, Invitrogen) staining for 1 hr and finally, cells were counterstained with a CellMask Deep Red (1/1000, H32721, ThermoFisher). Cells were imaged using the Yokogawa CellVoyager CV8000 (×20 objective) and analysis was done using Signals Image Artist.

Alternatively, ASC speck formation was assessed as described before (Fernandes-Alnemri et al, 2010). Briefly, $2 \times 10^6$ BMDMs in a six-well plate were collected and lysed in 0.5 ml lysis buffer (20 mM HEPES pH 7.5, 150 mM KCl, 1% NP-40, 0.1 mM PMSF and complete protease inhibitor) on ice by syringing 10x. The lysates were centrifuged at 6000 rpm at 4 °C for 10 min, pellets washed with PBS and resuspended in 0.5 ml PBS. Pellets were cross-linked with fresh disuccinimidyl suberate (DSS, 2 mM) for 30 min at RT and pellets centrifugated at 6000 rpm for 10 min. Cross-linked pellets were resuspended in 30 µl Laemmli buffer and separated by SDS-PAGE followed by transfer to a PVDF membrane and detection with primary antibody against mouse ASC (1/1000, pAL177, AG-25B-0006-C100, Adipogen).

Further, undifferentiated THP-1 ASC-mCherry cells were visualized by Incucyte S3 (Sartorius) using ×10 lens and speck formation followed over time. Quantification was done using IncuCyte software.

## Cytokine analysis

Supernatants from stimulated cells were collected at indicated time points and MesoScale Discovery (MSD) technology was used to detect levels of IL-1β, IL-6, and TNF according to the manufacturer's protocol (K151A9H; MSD). For undifferentiated THP-1 and NLRP3 KO validation, Luminex assay for IL-18, IL-1β, IL-6 and TNF (EPX01A-10267-901; EPX01A-10224-901; EPX01A-10213-901; EPX01A-10223-901; Procartaplex, Thermo Fisher) was performed. For in vivo experiments murine IL-1β ELISA (SMLB00C; R&D) was performed according to manufacturer's protocol.

## Western blotting

For western blot analysis, BMDMs were seeded in a 12-well plate at $1 \times 10^6$ cells per ml and lysates collected at indicated time points using cell lysis buffer (20 mM Tris HCl (pH 7.4), 200 mM NaCl and 1% NP-40) and Laemmli buffer. Protein samples were boiled at 95 °C for 10 min and separated by SDS-PAGE followed by transfer to PVDF membranes. Blocking, incubation with antibody, and washing of the membranes were done in PBS supplemented with 0.05% Tween 20 (v/v) and 3% (w/v) non-fat dry milk. Immunoblots were incubated overnight with primary antibodies against caspase-1 (1/1000; AG-20B-0042-C100, Adipogen), Gsdmd (1/1000; #209845, #219800; abcam), IL-1β (1/2000; GTX74034, Genetex), NLRP3 (1/1000; AG-20B-0014-C100, Adipogen) and β-actin (1/5000; sc-47778 HRP; SantaCruz). Horseradish peroxidase-conjugated goat anti-mouse or anti-rabbit secondary antibody was used to detect proteins by enhanced chemiluminescence (1/5000; Pierce, Thermo).

## In vivo LPS challenge

Cohorts of female wild-type and NLRP3$^{-/-}$ C57BL/6N mice with age between 8-12 weeks were pretreated by oral gavage with either vehicle (PEG400), reference compound or test compound at indicated doses. After 30 min, mice were challenged with an intraperitoneal dose of 10 mg/kg LPS (E. coli serotype O111:B4, Sigma, L4130) and after 4 h, mice were terminally sedated and blood was collected for cytokine analysis and bio-analysis.

## In vivo CAPS model

Cohorts of age-matched A350V CreERT2+/− and A350V CreERT2−/− animals were treated by oral gavage with tamoxifen (Sigma, T5648) for 5 days. In addition, they were dosed with vehicle PEG400 or compound A for 9 consecutive days followed by sacrifice 24 h after last dose. Bodyweight was measured daily. Terminal bleeding was performed, and plasma used to determine cytokine levels (IL-1β, #102517; Quanterix), Luminex (IL-18: EPX01A-20618-901, G-CSF: EPX01A-26034-901, IP-10: EPX01A-26018-901, IL-6: EPX01A-20603-901) and ELISA (SAA, ab215090; Abcam), technology according to manufacturer's guidelines. Fresh blood was analyzed to determine neutrophil count using Sysmex technology. Livers, lungs and spleens were isolated and evaluated for pathological findings using H&E staining.

## Histopathology and histopathologic evaluation

Representative specimens from the liver, lung, and spleen were sampled, fixed in 10% neutral buffered formalin. Fixed specimens were trimmed and processed routinely through the paraffin embedding technique. The embedded tissues were sectioned and stained with hematoxylin-eosin (H&E). Histopathological examination was performed by a board-certified toxicologic pathologist, and the microscopic findings were either graded (0: no finding, 1: minimal histological change, 2: slight, 3: moderate, 4: marked and 5: severe/massive histological change) or indicated as present without a grade.

## Statistical analysis

Independent experimental replicate numbers, statistical tests and $p$ values (considered statistically significant if $P < 0.05$) are described in the figure legends. For all statistical analyses, GraphPad Prism 10.0

### The paper explained

**Problem**

In response to internal or external triggers, inflammasomes are complexes that form in the cytoplasm. Among these inflammasomes, NLRP3 is the most thoroughly studied due to its involvement in neurodegenerative and chronic auto-inflammatory diseases. Unfortunately, the lack of potential NLRP3 clinical candidates results in the clinical field for NLRP3 inhibition remaining an unmet need.

**Results**

In this manuscript, we describe a novel chemical class with high selectivity for NLRP3, which has the potential to mitigate side effects and toxicities associated with conventional diaryl sulfonylurea-containing compounds. Moreover, the lead compound A shows higher activity against the auto-inflammatory NLRP3 Cryopyrin-Associated Periodic Syndrome (CAPS) mutants compared to conventional diaryl sulfonylurea-containing compound (MCC950).

**Impact**

This manuscript identifies a new small-molecule series, unique from the MCC950-like diaryl sulfonylurea-containing compounds, that can potently inhibit wild-type and clinically relevant NLRP3 mutants. Overall, this study identifies a new avenue for safe and better therapeutics for patients with NLRP3-driven auto-inflammatory diseases.

was used. Log-transformed cytokine data are used to fulfill the assumption of a normal distribution allowing the use of a parametric analysis. Inclusion/exclusion criteria were pre-established and no data were excluded based on experimental outcomes.

## Data availability

The coordinates and EM maps generated in this study have been deposited in the Protein Data Bank and the Electron Microscopy Data Bank under accession codes: PDB ID 9DH3, EMD-46855 (MBP-NLRP3ΔPYD/ + ADP/+Compound C). Source data are provided with this paper.

The source data of this paper are collected in the following database record: biostudies:S-SCDT-10_1038-S44321-024-00181-4.

## Peer review information

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

## Acknowledgements

The authors thank the screening group of Charles River Laboratories Beerse for compound testing, In Vivo Sciences for in vivo support and the Janssen patent department. The cryo-EM data were collected at NanoImaging Services (San Diego, CA). This study is funded by Johnson & Johnson R&D unit.

## Author contributions

**Rosalie Matico**: Conceptualization; Data curation; Formal analysis; Validation; Investigation; Visualization; Methodology; Writing—original draft; Writing—

review and editing. **Karolien Grauwen**: Data curation; Formal analysis; Validation; Investigation. **Dhruv Chauhan**: Conceptualization; Data curation; Formal analysis; Validation; Investigation; Visualization; Methodology; Writing —original draft; Writing—review and editing. **Xiaodi Yu**: Conceptualization; Data curation; Formal analysis; Validation; Investigation; Visualization; Methodology; Writing—original draft; Writing—review and editing. **Irini Abdiaj**: Conceptualization; Data curation; Formal analysis; Validation; Investigation; Visualization; Methodology; Writing—original draft; Writing— review and editing. **Suraj Adhikary**: Data curation; Formal analysis; Validation; Investigation; Methodology. **Ine Adriaensen**: Data curation; Formal analysis; Validation; Investigation. **Garcia Molina Aranzazu**: Data curation; Formal analysis; Validation; Investigation. **Jesus Alcázar**: Data curation; Formal analysis; Validation; Investigation. **Michela Bassi**: Data curation; Formal analysis; Validation; Investigation. **Ellen Brisse**: Conceptualization; Data curation; Formal analysis; Validation; Investigation; Methodology. **Santiago Cañellas**: Conceptualization; Data curation; Formal analysis; Validation; Investigation; Methodology. **Shubhra Chaudhuri**: Data curation; Formal analysis; Validation; Investigation. **Francisca Delgado**: Data curation; Formal analysis; Validation; Investigation. **Alejandro Diéguez-Vázquez**: Conceptualization; Data curation; Formal analysis; Validation; Investigation; Visualization; Methodology; Writing—original draft; Writing—review and editing. **Marc Du Jardin**: Conceptualization; Data curation; Formal analysis; Validation; Investigation. **Victoria Eastham**: Conceptualization; Data curation; Formal analysis; Validation; Investigation; Methodology. **Michael Finley**: Conceptualization; Data curation; Formal analysis; Validation; Investigation; Methodology. **Tom Jacobs**: Conceptualization; Data curation; Formal analysis; Validation; Investigation; Methodology. **Ken Keustermans**: Conceptualization; Data curation; Formal analysis; Validation; Investigation; Methodology. **Robert Kuhn**: Data curation; Formal analysis; Validation; Investigation. **Josep Llaveria**: Data curation; Formal analysis; Validation; Investigation. **Jos Leenaerts**: Data curation; Formal analysis; Validation; Investigation. **Maria Lourdes Linares**: Data curation; Formal analysis; Validation; Investigation. **Maria Luz Martín**: Data curation; Formal analysis; Validation; Investigation. **Rosa Martín-Pérez**: Data curation; Formal analysis; Supervision; Investigation; Methodology; Writing—review and editing. **Carlos Martínez**: Data curation; Formal analysis; Validation; Investigation. **Robyn Miller**: Data curation; Formal analysis; Validation; Investigation. **Frances M Muños**: Conceptualization; Data curation; Formal analysis; Validation; Investigation; Methodology. **Michael E Muratore**: Data curation; Formal analysis; Validation; Investigation. **Amber Nooyens**: Conceptualization; Data curation; Formal analysis; Validation; Investigation; Methodology. **Laura Perez-Benito**: Conceptualization; Data curation; Formal analysis; Validation; Investigation; Methodology. **Mathieu Perrier**: Conceptualization; Data curation; Formal analysis; Validation; Investigation; Methodology. **Beth Pietrak**: Conceptualization; Data curation; Formal analysis; Validation; Investigation; Methodology. **Jef Serré**: Conceptualization; Data curation; Formal analysis; Validation; Investigation; Methodology. **Sujata Sharma**: Conceptualization; Data curation; Formal analysis; Validation; Investigation; Methodology. **Marijke Somers**: Conceptualization; Formal analysis; Validation; Investigation; Methodology. **Javier Suarez**: Conceptualization; Formal analysis; Validation; Investigation; Methodology. **Gary Tresadern**: Conceptualization; Formal analysis; Validation; Investigation; Methodology. **Andres A Trabanco**: Conceptualization; Formal analysis; Validation; Investigation; Methodology. **Dries Van den Bulck**: Conceptualization; Formal analysis; Validation; Investigation; Methodology. **Michiel Van Gool**: Conceptualization; Data curation; Formal analysis; Validation; Investigation; Methodology. **Filip Van Hauwermeiren**: Conceptualization; Formal analysis; Validation; Investigation; Methodology. **Teena Varghese**: Formal analysis; Validation; Investigation; Methodology. **Juan Antonio Vega**: Data curation; Formal analysis; Validation; Investigation. **Sameh A Youssef**: Data curation; Formal analysis; Validation; Investigation. **Matthew J Edwards**: Conceptualization; Data curation; Supervision; Validation; Methodology; Project administration; Writing—review and editing. **Daniel Oehlrich**: Conceptualization; Data curation; Formal analysis; Supervision; Validation; Investigation; Visualization; Methodology; Writing—original draft; Project administration; Writing—review and editing. **Nina Van Opdenbosch**: Conceptualization; Data curation; Formal analysis; Supervision; Validation; Investigation; Visualization; Methodology; Writing—original draft; Project administration; Writing—review and editing; designed experiments.

Source data underlying figure panels in this paper may have individual authorship assigned. Where available, figure panel/source data authorship is listed in the following database record: biostudies:S-SCDT-10_1038-S44321-024-00181-4.

## Disclosure and competing interests statement

All authors are (or were) J&J employees when participating in this work and declare no competing interests.

# Expanded View Figures

**Figure EV1.   Structural insights into NLRP3 small-molecule comparison and proposed Compound A binding mode.**

(A) Schematic representation of NLRP3 architecture, with domains colored as follows: PYD (red), NBD (orange), HD1 (slate), WHD (light green), HD2 (dark green), and LRR (light gray). (B) NLRP3 small-molecule inhibitors were aligned using the central amine moiety as the reference point. The PDB IDs for each small-molecule-bound NLRP3 structure were shown. (C) Key interacting residues of NLRP3 with the small molecule are highlighted and color-coded as in Fig. EV1A. Note that the residues are grouped by subdomains but may not precisely represent the actual spatial interactions with NLRP3. Interacting residues of NLRP3 with compound A, observed in the simulation (Frame 50), are highlighted with dashed square. (D) Pairwise structural comparison of small-molecule binding to NLRP3. The small molecules are depicted as sticks in pink or yellow, while NLRP3 is shown as a cartoon, color-coded as in Fig. EV1A. Key interacting residue side chains are represented as lines, with colors corresponding to their respective subdomains. (E) The compound-accessible surface area was plotted against the NLRP3 area changes caused by compound binding (in $Å^2$). These areas and their differences were calculated using the AreaMol program from the CCP4 package. Values derived from real complex structures are shown in black spheres, while those from simulations (Frame 50, 500, and 1000) are represented in pink and yellow spheres for compounds C and A, respectively. (F) Violin plot showing distances between Val353 and Glu629, and between compounds A or C and Arg578 (a) or Glu636 (b) from the entire simulation set. Insets display the measured distances between compounds A or C and Arg578 (a) or Glu636 (b). Simulations for Apo, Compound A, and C are colored gray, yellow, and magenta, respectively. Slate and red bars indicate the distances of Val353 and Glu629 from NLRP3's closed (PDB IDs: 7PZC, 7VTP, 7VTQ, 8SWK, 8SXN, 8ETR, 7ALV, 8WSM, 9DH3, and 8RI2) and open structures (PDB IDs: 8SWF and 8EJ4). MD simulations were conducted three times ($n = 3$).

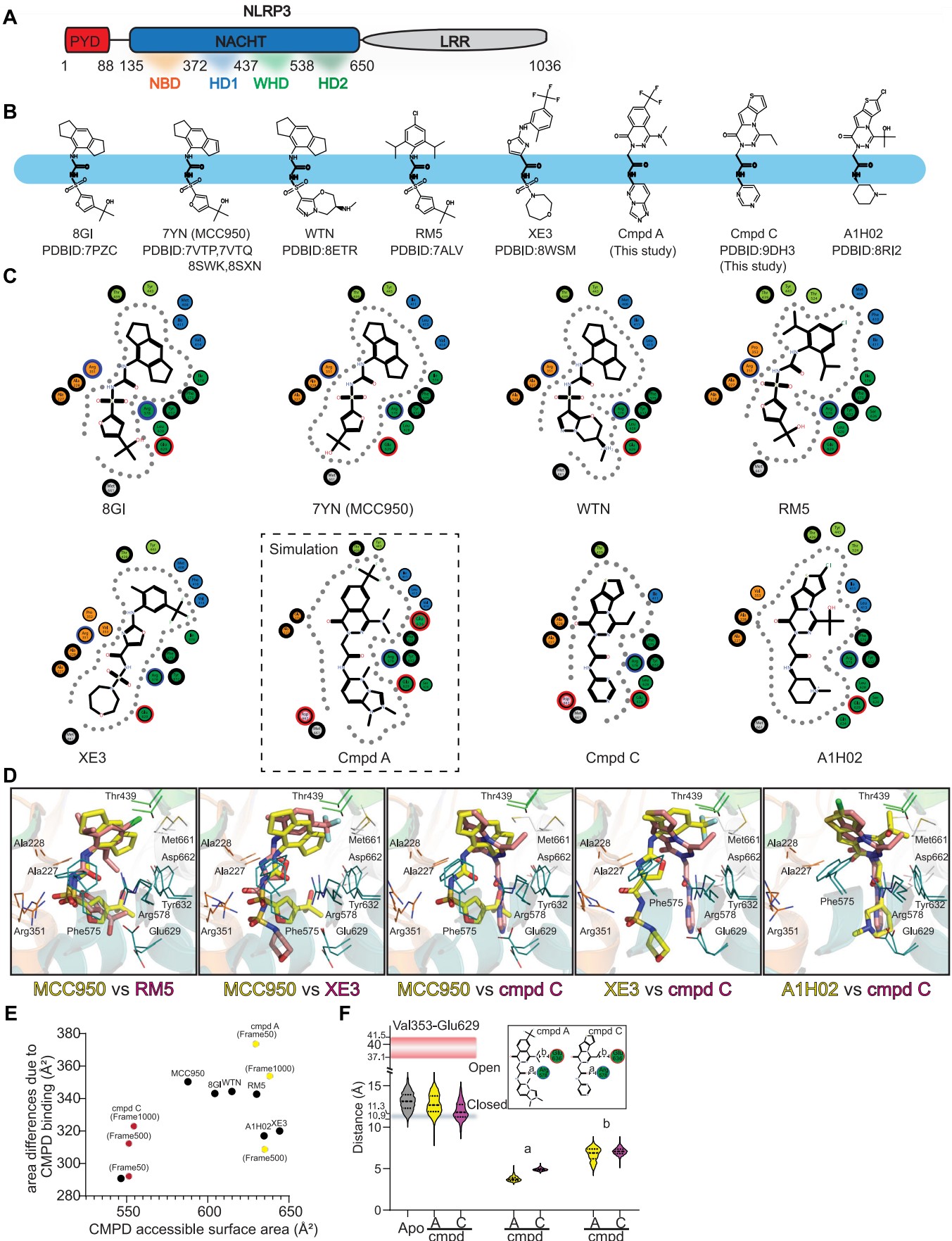

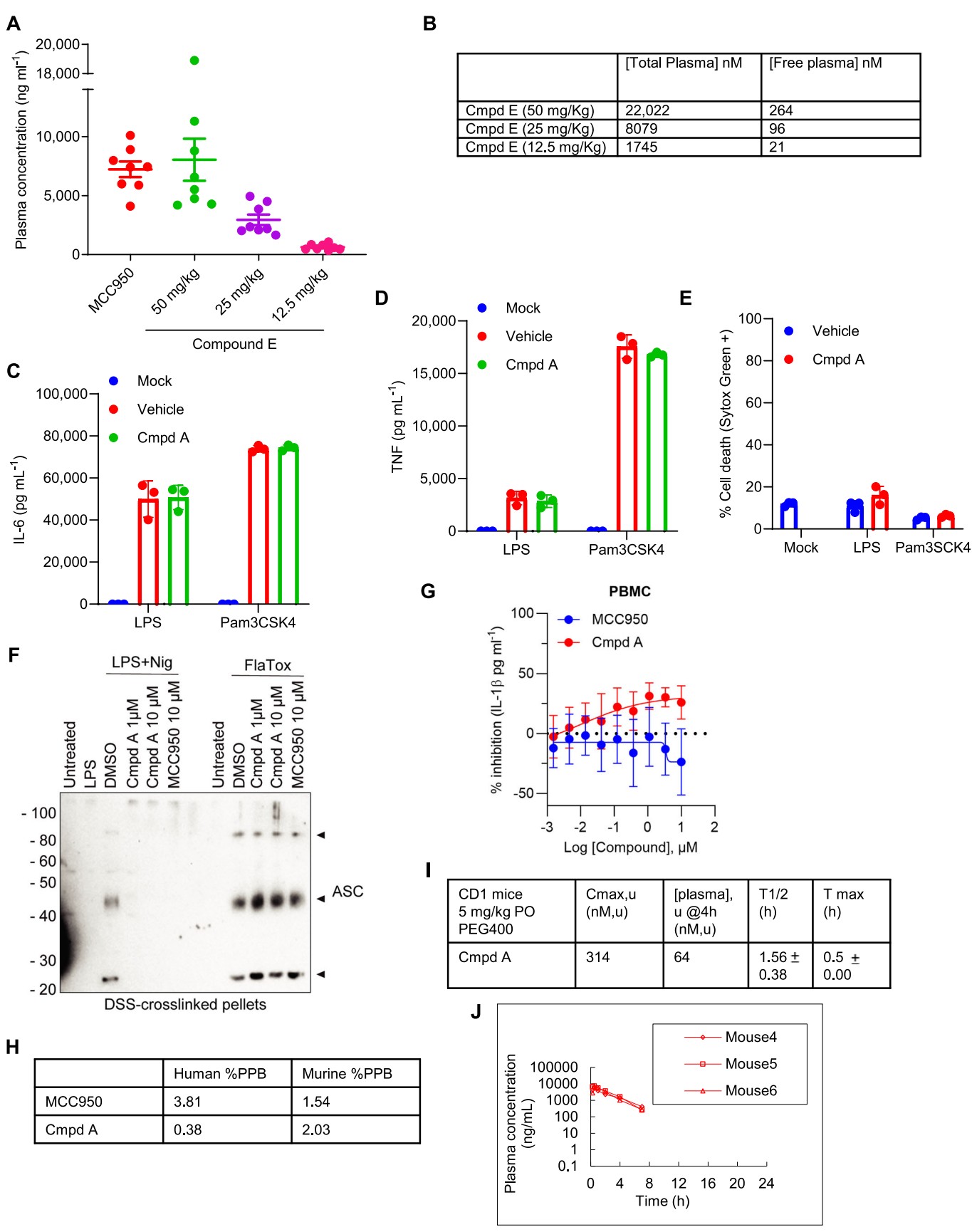

**Figure EV2. Characterization of compound E and A.**

(A, B) C57BL/6 mice were pretreated with reference compound MCC950 or compound E for 30 min followed by LPS exposure for 4 h. Blood collected and compound concentration determined in plasma. Calculation of free compound levels based on plasma-protein binding. Per group 8 animals were used and mean +/- SEM is depicted. Free plasma levels are calculated based on protein binding. (C, D) LPS or Pam3CSK4-induced NFκB led to the production of IL-6 (C) and TNF (D) which was not impacted by pretreatment with compound A. Representative image from $n = 3$ shown with 3 technical repeats and mean $+/-$ SD depicted. (E) Cell death induced by 16 h treatment with LPS or Pam3CSK4 in combination with compound A. Representative image from $n = 3$ shown with 3 technical repeats and mean $+/-$ SD depicted. (F) Wild-type BMDMs either primed with LPS or left untreated for 2 h in presence of Cmpd A (1–10 μM) or MCC950 (10 μM) were stimulated with Nigericin (Nig) or FlaTox for 2 h. DSS-crosslinking of lysates was performed after stimulation and high-order oligomerisation of ASC was detected by immunoblotting. Western blot is representative image of two independent experiments ($n = 2$). (G) Pam3csk4-primed human PBMCs were treated with reference compound and compound A for 30 min followed by NdlTox stimulation for 3 h. Supernatant is collected and used for cytokine detection on MSD. PBMCs from 3 independent donors ($n = 3$) were used quadruplicate and all data pooled. Mean $+/-$ SD is depicted. (H) % plasma-protein binding of reference compound and compound A. (I, J) Pharmacodynamic characteristics of compound A determined in CD1 mice.

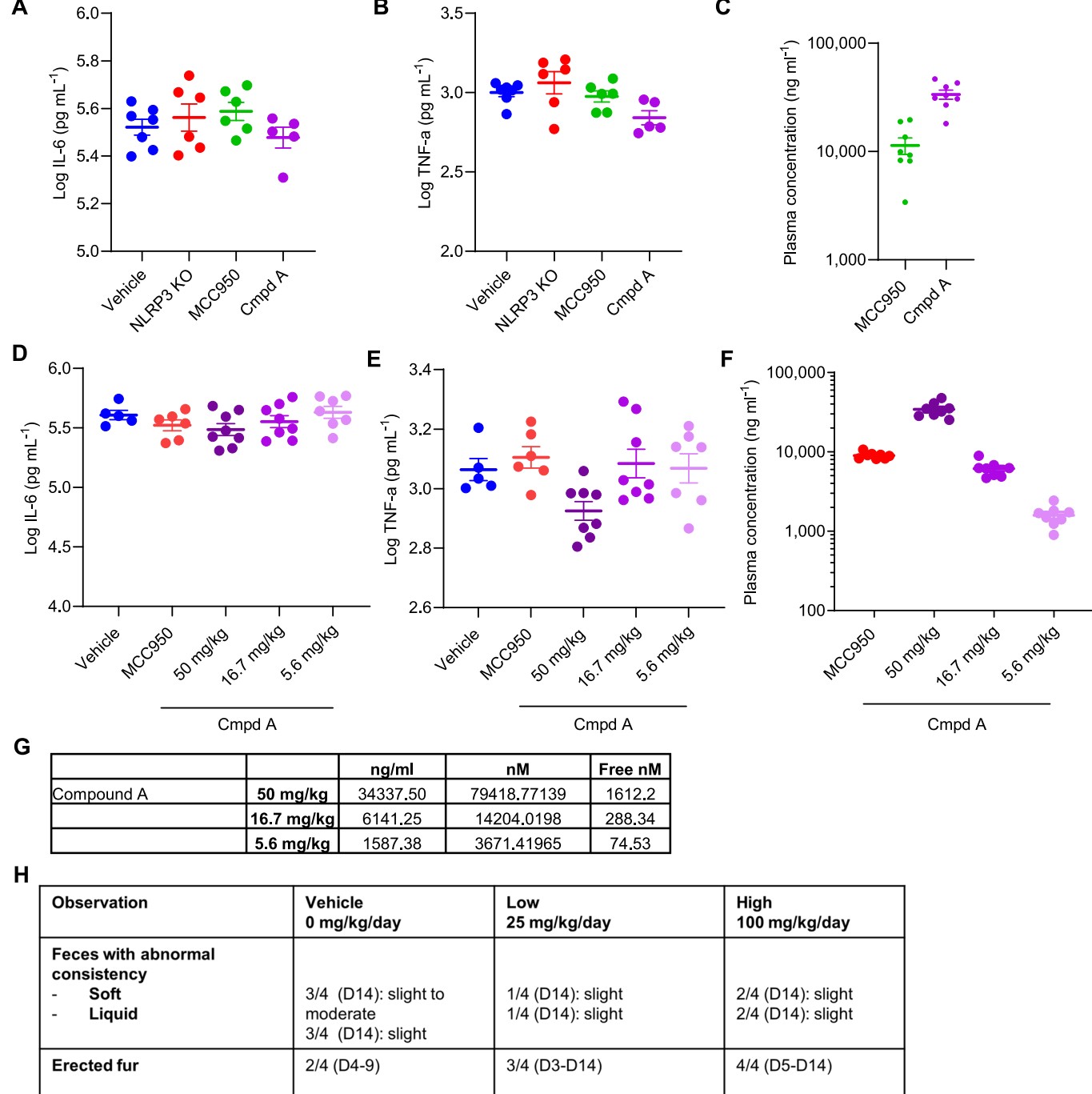

| Compound A | | ng/ml | nM | Free nM |
|---|---|---|---|---|
| Compound A | **50 mg/kg** | 34337.50 | 79418.77139 | 1612.2 |
| | **16.7 mg/kg** | 6141.25 | 14204.0198 | 288.34 |
| | **5.6 mg/kg** | 1587.38 | 3671.41965 | 74.53 |

**H**

| Observation | Vehicle 0 mg/kg/day | Low 25 mg/kg/day | High 100 mg/kg/day |
|---|---|---|---|
| **Feces with abnormal consistency**<br>- **Soft**<br>- **Liquid** | 3/4 (D14): slight to moderate<br>3/4 (D14): slight | 1/4 (D14): slight<br>1/4 (D14): slight | 2/4 (D14): slight<br>2/4 (D14): slight |
| **Erected fur** | 2/4 (D4-9) | 3/4 (D3-D14) | 4/4 (D5-D14) |
| **Generalized oily fur aspect** | - | | 1/4 (D3-D9) |
| **Color discharge, mouth** | - | 1/4 (D10) | - |
| **Histopathology analysis of organs (heart, kidney, liver, spleen)** | 4/4: normal heart<br>4/4: normal kidney<br>4/4: normal liver<br>4/4: normal spleen | 4/4: normal heart<br>4/4: normal kidney<br>4/4: normal liver<br>4/4: normal spleen | 4/4: normal heart<br>4/4: normal kidney<br>4/4: normal liver<br>4/4: normal spleen |

Soft to liquid feces was observed in all groups including control on Day 14.
These signs are commonly seen in mice dosed orally with PEG400 and were thus considered vehicle-related.
-: no noteworthy observation

◀ **Figure EV3.  In vivo validation of compound A.**

(**A, B**) Levels of IL-6 and TNF determined in C57BL/6 mice treated with compound followed by LPS injection for 4 h. Per group 8 animals were used and mean +/− SEM is depicted. (**C**) Compound levels determined in C57BL/6 mice treated with compound followed by LPS injection for 4 h. Per group 8 animals were used and mean +/− SEM is depicted. (**D, E**) Levels of IL-6 and TNF determined in C57BL/6 mice treated with different doses of compound followed by LPS injection for 4 h. Per group 8 animals were used and mean +/− SEM is depicted. (**F, G**) Compound levels determined in C57BL/6 mice treated with compound followed by LPS injection for 4 h. Calculation of free compound levels at endpoint. Per group 8 animals were used and mean +/− SEM is depicted. (**H**) Toxicology profile of compound A after a preclinical short 14 d tox study in mice. Per group, 4 animals are used.

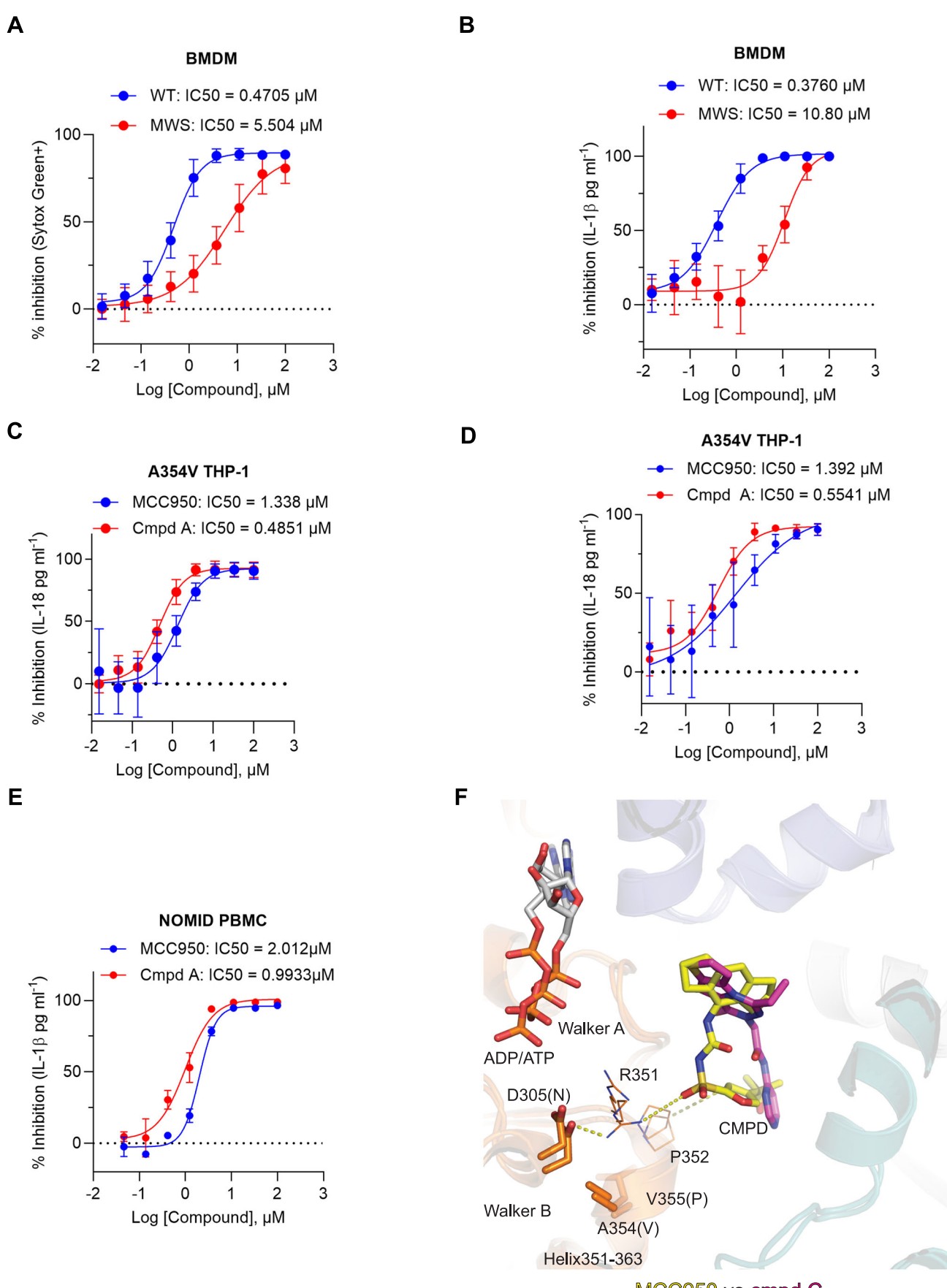

◀ **Figure EV4. Impact of CAPS mutations on the efficacy of compound A.**

(A, B) Wild-type or conditional MWS BMDMs were treated with a dose response of MCC950 and effect on cell death (A) and IL-1β release (B) was determined. BMDMs from 3 independent animals ($n = 3$) were used in duplicate and pooled data depicted as mean $+/-$ SD. (C, D) Undifferentiated THP-1 cells were treated with doxycycline to allow expression of A354V mutant NLRP3 whereafter cells were treated with reference compound or compound A in dose response followed by treatment with LPS (A) or LPS + Nig (B). Supernatant was collected and IL-18 detected using Luminex. Two independent repeats ($n = 2$) were performed in quadruplicate and pooled data depicted as mean $+/-$ SD. (E) PBMCs from a patient with NOMID mutation were treated with reference compound or compound A followed by LPS + Nig. Supernatant was collected and IL-1β measured by MSD. PBMCs were treated in duplicate, and data represented as mean $+/-$ SD. (F) Comparison of MCC950 and compound C binding to human NLRP3. Small molecules are shown as pink (compound C) and yellow (MCC950) sticks, while NLRP3 is depicted as a color-coded cartoon, as in Fig. EV1A. Side chains of Arg351 and Pro352 are displayed as lines, and mutations from CAPS (Ala354Val), FCAS (Leu355Pro), and NOMID (Asp305Asn) patients are represented as sticks, colored according to their respective subdomains. Yellow dashed lines indicate potential interactions between residues and MCC950.

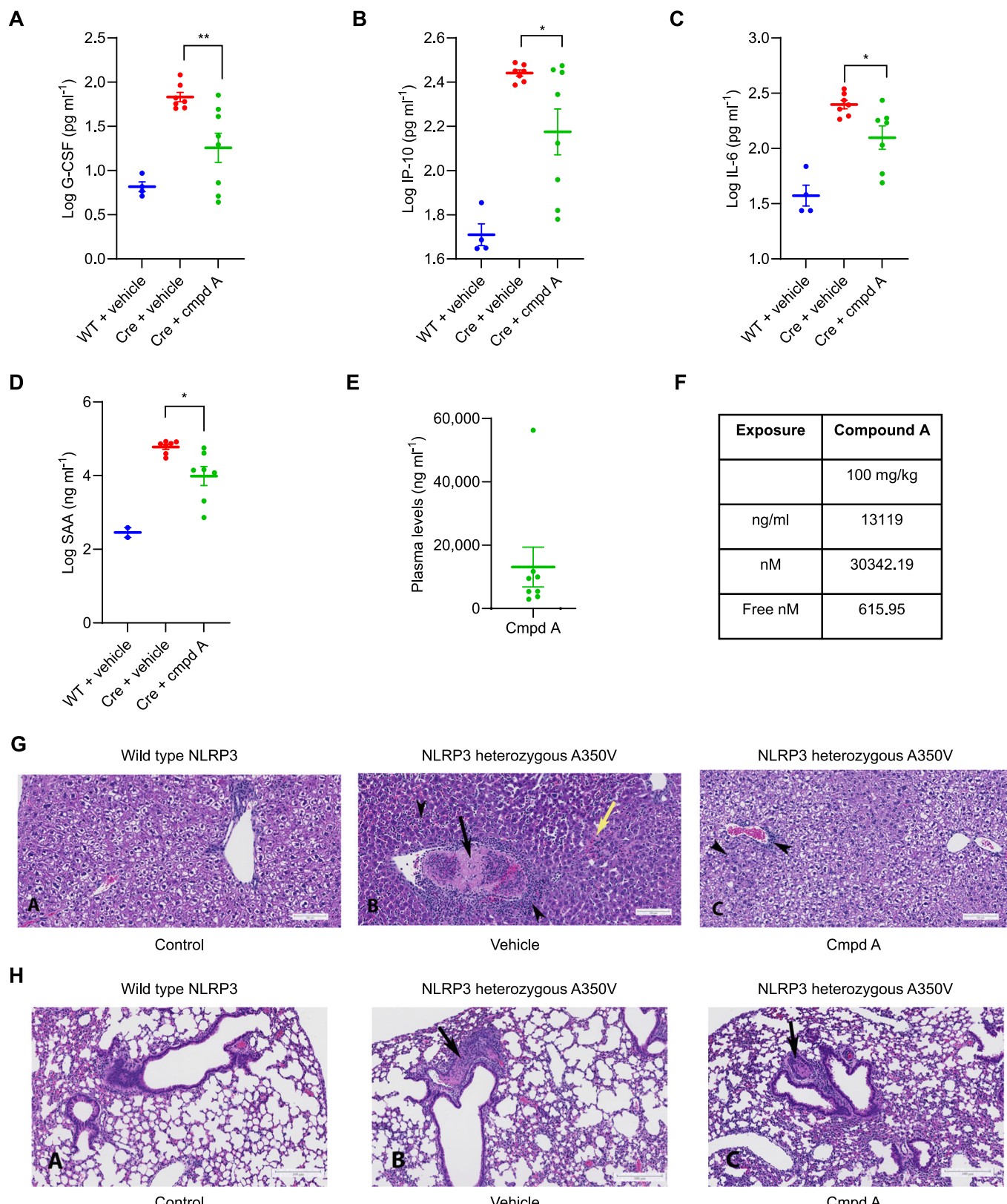

◀ **Figure EV5. In vivo validation of compound A using the MWS mouse model.**

(A–D) Tamoxifen-induced MWS-mediated disease was measured using G-CSF (A), IP-10 (B), IL-6 (C) and SAA (D) in plasma after collection of the blood on day 10. Per group 4 (WT vehicle), 7 (Cre + vehicle) or 8 (Cre + cmpd A) animals were used and mean $+/-$ SEM is depicted. Unpaired two-tailed t-test with welch's correction in Cre treated vs vehicle samples was performed: **$P = 0.0058$ (G-CSF), *$P = 0.0317$ (IP-10), *$P = 0.0145$ (IL-6), *$P = 0.0145$ (SAA). (E, F) Levels of compound in plasma in blood collected at day 10, 24 h after last oral dosing. Eight animals were used in this group and mean $+/-$ SEM is depicted. Free compound was calculated (F). (G) Photomicrograph (H&E) of liver sections from a control wildtype (A), vehicle-treated NLRP3 heterozygous (B) and compound A-treated NLRP3 heterozygous (c) mouse. Note the hepatocellular necrosis (yellow arrow), portal vein thrombosis (black arrow), and inflammatory cells infiltration in the sinusoids and portal triads (arrowheads) of the vehicle-treated liver. In contrast, compound A-treated liver appeared histologically comparable to wild-type liver and exhibited only minimal inflammatory cells infiltration (arrowheads) and significant reduction in the severity of thrombosis and necrosis. Bar$=$ 100 microns. (H) Photomicrograph (H&E) of lung sections from a control wildtype (A), vehicle-treated NLRP3 heterozygous (B) and compound A-treated NLRP3 heterozygous (c) mouse. Compound A-treated lung appeared histologically comparable to wild-type liver and exhibited reduction in the severity of thrombosis (arrows) when compared to vehicle-treated NLRP3 heterozygous. Bar $=$ 100 microns.

