## [Peer Review File · EMBO Molecular Medicine]

Navigating from phenotypic screen to clinical candidate: selective targeting the NLRP3 inflammasome

Rosalie Matico, Karolien Grauwen, Dhruv Chauhan, Xiaodi Yu, Iriñi Abdiaj, Suraj Adhikary, Ine Adriaensen, Garcia Aranzazu, Jesus Alcázar, Michela Bassi, Ellen Brisse, Santiago Cañellas, Shubhra Chaudhuri, Francisca Delgado, Alejandro Diéguez-Vázquez, Marc Du Jardin, Victoria Eastham, Michael Finley, Tom Jacobs, Ken Keustermans, Robert Kuhn, Josep Llaveria, Jos Leenaerts, Maria Linares, Maria Martín, Rosa Martín-Pérez, Carlos Martínez, Robyn Miller, Frances Muños, Michael Muratore, Amber Nooyens, Laura Perez, Mathieu Perrier, Beth Pietrak, Jef Serré, Sujata Sharma, Marijke Somers, Javier Suarez, Gary Tresadern, Andres Trabanco, Dries Van den Bulck, Michiel Van Gool, Filip Van Hauwermeiren, Teena Varghese, Juan Vega, Sameh Youssef, Matthew Edwards, Daniel Oehlrich, and Nina Van Opdenbosch

Corresponding author: Nina Van Opdenbosch (nvanopde@its.jnj.com)

Review Timeline:

Submission Date:	23rd May 24
Editorial Decision:	22nd Jun 24
Revision Received:	26th Sep 24
Editorial Decision:	16th Oct 24
Revision Received:	23rd Nov 24
Accepted:	26th Nov 24

Editor: Zeljko Durdevic

Transaction Report:

22nd Jun 2024

Dear Dr. Van Opdenbosch,

Thank you for the submission of your manuscript to EMBO Molecular Medicine. We have now received feedback from the three reviewers who agreed to evaluate your manuscript. All three referees recognize interest of the study but also raise important and partially overlapping concerns that should be addressed in a major revision. Particular attention should be given to the better characterization of the safety and toxicity of the compound, the direct comparison to MCC950-derivatives and better highlighting the conceptual advance of the study. Investigation of the effects of compound A in other NLRP3-related disease model as suggested by the referee #2 is welcome and would indeed strengthen the manuscript further. However, this is not essential for further consideration of the manuscript. If you would like to discuss further the points raised by the referees, I am available to do so via email or video. Let me know if you are interested in this option.

We would welcome the submission of a revised version within three months for further consideration. Please let us know if you require longer to complete the revision.

I look forward to receiving your revised manuscript.

Yours sincerely,

Zeljko Durdevic

We require:

- 1) A .docx formatted version of the manuscript text (including legends for main figures, EV figures and tables). Please make sure that the changes are highlighted to be clearly visible.
- 2) Individual production quality figure files as .eps, .tif, .jpg (one file per figure). For guidance, download the 'Figure Guide PDF': (<https://www.embopress.org/page/journal/17574684/authorguide#figureformat>).
- 3) A .docx formatted letter INCLUDING the reviewers' reports and your detailed point-by-point responses to their comments. As part of the EMBO Press transparent editorial process, the point-by-point response is part of the Review Process File (RPF), which will be published alongside your paper.
- 4) A complete author checklist, which you can download from our author guidelines (<https://www.embopress.org/page/journal/17574684/authorguide#submissionofrevisions>). Please insert information in the checklist that is also reflected in the manuscript. The completed author checklist will also be part of the RPF.
- 5) Please note that all corresponding authors are required to supply an ORCID ID for their name upon submission of a revised

manuscript.

6) It is mandatory to include a 'Data Availability' section after the Materials and Methods. Before submitting your revision, primary datasets produced in this study need to be deposited in an appropriate public database, and the accession numbers and database listed under 'Data Availability'. Please remember to provide a reviewer password if the datasets are not yet public (see <https://www.embopress.org/page/journal/17574684/authorguide#dataavailability>).

13) Author contributions: You will be asked to provide CRediT (Contributor Role Taxonomy) terms in the submission system. These replace a narrative author contribution section in the manuscript.

14) A Conflict of Interest statement should be provided in the main text.

Please also suggest a striking image or visual abstract to illustrate your article as a PNG file 550 px wide x 300-800 px high.

***** Reviewer's comments *****

Referee #1 (Comments on Novelty/Model System for Author):

This work characterizes a new NLRP3 inflammasome inhibitor, which is highly specific and potent. The work is interesting, but it needs to address a few important concerns that are indicated below in the remarks to the authors section.

Referee #1 (Remarks for Author):

In the study, Matico R et al., have performed a high throughput screening for the identification of an NLRP3-selective small molecule inhibitor (in this case, it was indicated as compound A). The lead was optimized to overcome the suboptimal compound profile through careful analysis. The authors have mentioned other contemporary studies that reported compounds similar to the ones identified in this study, but the screening pipeline designed by the authors seems robust. Experiments performed to assess the efficacy of the compound in mouse/human cell lines or in vivo show an overall higher potency of the compound than MCC950. However, the issue regarding the safety/toxicity of the compound has not been adequately addressed in the manuscript. I have a few critical comments. Please see below

1. In the introduction, the organization of information must be extensively edited for better readability. It is important to highlight whether all previously identified pharmacophores belong to the diaryl sulfonylurea compound class. The explanation for why there is a need for a new compound when there are existing effective compounds under clinical trials is not convincing. It can be more clearly outlined.
2. Page 6: What is the identified cluster 1? This needs to be outlined in more detail. A brief introduction about the chemical class of the compounds B, C, E, etc must be provided at an initial stage for better understanding of the reader.
3. Why was the PYD construct/structure used for the cryo-EM studies? I assume that the authors used this because of the predicted specificity of the compound to the NATCH domain binding. However, it is important to check if the proposed compounds do not target PYD. The overlay between the compound C bound MBP-hNLRP3- Δ PYD and the closed NLRP3 (PDBID: 7VTP) protomer structures have not been shown.
4. The authors might have used the 20X objective of the Incucyte S3 imaging system for the ASC speck formation experiment. At 20X magnification, the visualisation and quantification of ASC specks in the stable THP-1 cells might be error-prone. There are no representative images for the reported data. Additionally, it is recommended to use confocal imaging at 60X or 100X to validate the presented data.
5. The data shown in Suppl Fig 11 is not representative of the histopathology findings reported in the manuscript text in the rodent 14-day mini-tox experiment. Since the authors have claimed that compound A is safer in comparison to the existing inhibitors (especially MCC950), there should be a comparative analysis of the toxicity levels between compound A and MCC950. In addition to this, there should be representative data showing the effects of compound A on organs examined (heart, kidney, liver, and spleen). A robust cytotoxicity analysis for compound A at the potent concentration should be performed at least in the mentioned cell lines.
6. In supplementary fig 5, panel C has not been annotated.

Referee #2 (Remarks for Author):

In this paper, Matico R et al. screened a series of compounds and identified compound A as a novel chemical class of NLRP3

inhibitor, which possesses high selectivity and potency both in vivo and in vitro. More impressively, the compounds deprived from the class inhibitors also exhibited high efficacy in addressing CAPS mutations without inducing toxic side effects. Overall, the findings are novel and interesting, providing priming candidates for the NLRP3-relevant disorders, such as CAPS. However, the following concerns and questions should be addressed to further strengthen the study.

Major comments:

1. The authors screened a novel chemical class cluster which could suppress NLRP3 inflammasome activation. Whether the class have a common feature, such as structural similarity? The authors should explore or discuss it.
2. The authors showed the compound A-H inhibited NLRP3 activation in different degrees. Why they exhibited the different role in the progress? Additionally, were compound A-H represented the class inhibitors and screened out casually?
3. It has been reported a same chemical class of highly potent and selective NLRP3-targeted inhibitors which reveals a well-defined molecular mechanism to complement existing MCC950-based NLRP3 inhibitors in pharmacological studies (doi: 10.26508/lsa.202402644), and lots of findings in the paper are similar to it, which greatly limits the innovation of the study.
4. In Fig.3E, the inhibition of dose of Compound E in IL-1 β secretion seems almost indistinctive, which was controversial with the conclusion of the authors.
5. In Fig.2 and Suppl Fig.2, why the authors used NLRP3- Δ PYD, but not the full-length NLRP3, to explore the interaction of compound and NLRP3?
6. MCC950 interacts the WALKER B motif of NLRP3, blocks ATP hydrolysis and inflammasome formation, thereby inhibiting inflammasome activation (doi: 10.1038/s41589-019-0277-7). The authors showed compound C and MCC950 both bind to the same pocket of NLRP3 but interact with the protein differently. Whether compound C also blocks ATP hydrolysis and inflammasome formation. Whether compound A bind to the pocket?
7. Fig.1F has no data showing the NLRP3 inhibition of compound C and D, just their chemical structural formula. Fig. 1G lacks the data related to IC50 for TNF- α and IL-6 secretion.
8. The authors should simultaneously detect both original and reconstructive metabolic stability and solubility of compound A in Fig.3D to highlight the role of further optimization.
9. Fig.4K lacked the related data to demonstrate the inhibition of compound A in IL-1 β secretion.
10. The biological effects of compound A in other NLRP3-related disease model, such as DSS-induced colitis, should be investigated
11. Suppl Fig.13G lacked the data related lung and spleens of in vivo experiment (Page13) .

Minor comments:

1. The name of inhibitor in figure legend (compound JNJ-666) of Fig.1D was not in line with the name in Figure (compound B).
2. The abbreviations are usually used defined at the first use in the abstract as well as in the main text, such as Poc or RHS in the 3rd part of Results.
3. In the whole part of Result, the author should make the conclusion in line with the corresponding Figures, such as the findings related NLRP3-KO should be described as Fig.5A behind the conclusion.
4. The final sentence in Result section is summary, but the authors added a reference (20).
5. There are numerous mistakes in the manuscript (especially in Results section). Please have the text examined carefully.

Referee #3 (Remarks for Author):

NLRP3 activity is recognized to cause or fuel a number of highly prevalent diseases including metabolic diseases (diabetes, obesity), neurodegenerations, cardiovascular diseases, therefore its pharmacological inhibition has a high therapeutic potential. In this manuscript, Matico et al. described a novel family of compounds that specifically inhibits NLRP3 by direct targeting. Chemical optimization of the initial compounds led to the development of compound A with physico-chemical properties that allow in vivo efficiency with oral delivery in mouse, and ex vivo efficiency in human cell lines and primary cells. These compounds target the same pocket as MCC950 but with a different binding orientation. While usage of MCC950 and its derivatives in clinic may be jeopardized by their hepatotoxicity, compound A is apparently not hepatotoxic in mice, which would constitute a major advantage. This study is well conducted in its methodology and interpretation of the results. The main limitation is that 3 other independent studies recently described the same family of compounds as novel NLRP3 inhibitor, which may reduce the novelty of this study for publication in *Embo Molecular Medicine*. Nevertheless, this study is the most advanced concerning compound optimization for in vivo use.

Major points :

- Compounds with similar structure, with identical mode of action and binding site on NLRP3 was recently described by 3 other groups (PMID: 38519142, PMID: 37756547, PMID: 38175811). Although it confirms the robustness of the results, it may decrease the novelty of this manuscript. Nevertheless, this manuscript provides a better characterization of the physico, chemical and pharmacokinetic properties of the compounds.

- Lack of hepatotoxicity of this novel class of compounds is presented as one of the major advantages as compared to MCC950-derivatives. It would be critical to compare the 2 class of compounds in parallel to prove this point. Concerning reference supporting the hepatotoxicity of MCC950, the authors cite ref 20 (PMID: 37223529), which itself cites other reviews (<https://doi.org/10.1021/cen-09807-cover> and PMID: 33255820). It would be better that the authors cite the original study that relates original data supporting this claim. Alternatively, the authors should show that MCC950 causes hepatotoxicity in the same experimental settings as in suppl Fig13G for direct comparison, in order to demonstrate the superiority of compoundA over MCC950 described at the end of page15

Minor points:

- Compound B structure should be disclosed to allow comparison with other publication that recently characterized compounds of the same cluster (PMID: 38519142)
- Suppl Fig1: Description of the quantification methods, normalization and statistical analysis would be helpful
- Fig2E: It would be helpful to represent compdC and MCC950 structural localisation in exactly the same orientation, or with superimposed images to facilitate the comparison.
- Statistical tests, number of independent experiments and replicates should be indicated for all figures
- Fig3 panels and legend: abbreviations should be limited to the most standard ones, and indicated in the legend in full words. Units should be expressed with the standard abbreviations. For example, mpk should be modified to mg/kg to comply to standard abbreviations and units
- Fig4A-B,D-J: unprocessed data (in addition to the displayed % Inhibition) should be added in supplemental figures
- Fig5A and Supplemental Fig10: secretion of cytokines following LPS injection are highly dynamic in time. Plasma TNF usually peaks around 1h post-LPS injection. A kinetics analysis of IL-1b, TNF, IL6 at 1h, 2h, 4h is necessary to support the authors claims
- Page 14: the link between the screen methods published in ref 38, and the methods or results presented in the present manuscript is unclear. The authors may rephrase these few sentences to better explain the point they attempt to make.

Dear Dr Durdevic,

Firstly, we would like to thank all the reviewers for their valuable time and critical suggestions to
improve and strengthen this manuscript. Please find our point-to-point replies to the questions
below in blue color.

Referee #1 (Comments on Novelty/Model System for Author):

This work characterizes a new NLRP3 inflammasome inhibitor, which is highly specific and
potent. The work is interesting, but it needs to address a few important concerns that are
indicated below in the remarks to the authors section.

We thank the reviewer for the critical reading and have addressed the concerns below in this P2P.

Referee #1 (Remarks for Author):

In the study, Matico R et al., have performed a high throughput screening for the identification of
an NLRP3-selective small molecule inhibitor (in this case, it was indicated as compound A). The
lead was optimized to overcome the suboptimal compound profile through careful analysis. The
authors have mentioned other contemporary studies that reported compounds similar to the
ones identified in this study, but the screening pipeline designed by the authors seems robust.
Experiments performed to assess the efficacy of the compound in mouse/human cell lines or in
vivo show an overall higher potency of the compound than MCC950. However, the issue regarding
the safety/toxicity of the compound has not been adequately addressed in the manuscript. I have
a few critical comments. Please see below

1. In the introduction, the organization of information must be extensively edited for better
readability. It is important to highlight whether all previously identified pharmacophores belong
to the diaryl sulfonyleurea compound class. The explanation for why there is a need for a new
compound when there are existing effective compounds under clinical trials is not convincing. It
can be more clearly outlined.

We thank the reviewer for the comment. MCC950 biosimilars carry the potential for hepatotoxicity
as described in Shah *et al* (2015, PMID: 26206150). The competitors with distinct MoA as
described in the introduction, appear to be pathway inhibitors, which might result in additional
undesired off-target activities, thereby leaving the unmet need for structurally distinct chemical
matter that target the same MoA as MCC950 but lacks the hepatotoxicity. We have now indicated
structurally different molecules from MCC950 in the text to give more clarity to the readers.
Modifications were made to the text (lines 137-142) to clarify this.

2. Page 6: What is the identified cluster 1? This needs to be outlined in more detail. A brief
introduction about the chemical class of the compounds B, C, E, etc must be provided at an initial
stage for better understanding of the reader.

We thank the reviewer for this question. After the screening campaign, 27 distinct chemical
classes were identified as potential NLRP3 inhibitors. Using stringent triaging, described in more
detail in the text (lines 178-184), cluster 1 was identified as the main chemical class selectively
and potently acting on inhibiting the NLRP3 inflammasome. Cluster 1 compounds B – C and D

are tricyclic core compounds with a side chain amide (lines 182 – 183). To provide more structural
similarity, we have now added structure of compound B along with compound C and D (Fig. 1F).
Next, target engagement showed direct binding of compound C (section 2 in results) and an
extensive structure- activity relationship (section 3 in results) led to the identification of
compound A with desired properties.

3. Why was the Δ PYD construct/structure used for the cryo-EM studies? I assume that the authors
used this because of the predicted specificity of the compound to the NATCH domain binding.
However, it is important to check if the proposed compounds do not target PYD. The overlay
between the compound C bound MBP-hNLRP3- Δ PYD and the closed NLRP3 (PDBID: 7VTP)
protomer structures have not been shown.

Thank you for the insightful question and suggestion. At this stage of drug discovery, we have a
clear understanding of how compound C engages with the NACHT domain, and structural data
suggests potential for improving its binding potency. However, as the reviewer noted, we have not
yet examined whether the compound non-specifically binds to the PYD or other unknown cellular
partners, which could affect binding potency in cells versus *in vitro*. Our data shows the IC₅₀ for
ATPase inhibition of MBP-hNLRP3- Δ PYD (33 nM, Appendix Figure S6) (lines 235 – 236) closely
aligns with our cellular inhibition data for full-length NLRP3 (IL1 β , IC₅₀: 41 nM, Figure 1G),
suggesting minimal off-target effects. During the resubmission process, the authors also
attempted HDX and Cryo-EM studies on the full-length NLRP3 complexed with compound C,
however, were unsuccessful, likely due to challenges in handling full-length NLRP3 *in vitro*.

We have added an extended figure (Figure EV1) (lines 211- 221) that provides a pairwise
comparison of structurally enabled compound binding to NLRP3, also attached here for
convenience. In panel D (middle), the overlay of MCC950 (7YN) and compound C is shown.

**Figure 1: Structural insights into NLRP3 small molecule comparison and proposed**
 **Compound A binding mode.**

4. The authors might have used the 20X objective of the Incucyte S3 imaging system for the ASC
 speck formation experiment. At 20X magnification, the visualisation and quantification of ASC
 specks in the stable THP-1 cells might be error-prone. There are no representative images for the
 reported data. Additionally, it is recommended to use confocal imaging at 60X or 100X to validate
 the presented data.

We thank the reviewer for this comment. We used 10x magnification to visualize the ASC specks.
 As you can appreciate from the images below, diffuse ASC is not counted while the condensation
 of ASC into a speck is seen as a positive signal and counted (blue mask). Representative images
 show ASC in red and the counting mask in blue. Although the quantification with Incucyte
 software is exact, it is widely accepted as technique for quantification and the conclusion is very

robust and the dose-response highly reproducible. The aim of this experiment is to show the
differentiation between compound A-mediated effect on NLRP3 and NLRC4- induced ASC
specks which is clearly visible from the images below.

**Figure 2: representative image showing ASC in red and counting mask of analysis in blue.**

Figure 2 shows the masking used for the quantification of the ASC specks using the Incucyte
software.

**Figure 3: representative images from different concentrations of Cmpd A with Nigericin or**
**NdlTox treatment.**

Figure 3 shows the raw data comparing NLRP3 and NLRC4-treated cells in the presence of
compound A. The NdlTox treated wells are not affected by the compound treatment while in the
nigericin-treated wells, a nice dose response effect is seen. Quantification of the plate is shown
below in figure 4.

Figure 4: Incucyte images of compound A-treated cells stimulated with Nigericin with the quantification of ASC specks depicted below.

The raw data images are mirrored in the graphs below indicating the applied masking works well. Raw data for this figure is provided with the resubmission including the data shown above.

5. The data shown in Suppl Fig 11 is not representative of the histopathology findings reported in the manuscript text in the rodent 14-day mini-tox experiment. Since the authors have claimed that compound A is safer in comparison to the existing inhibitors (especially MCC950), there should be a comparative analysis of the toxicity levels between compound A and MCC950. In addition to this, there should be representative data showing the effects of compound A on organs examined (heart, kidney, liver, and spleen). A robust cytotoxicity analysis for compound A at the potent concentration should be performed at least in the mentioned cell lines.

We thank the reviewer for highlighting this. The representative table in Figure EV3H (old Suppl Fig 11) was updated to add the data described in the text and highlight the organs examined in our 14d mini-tox experiment (lines 357-360).

Additionally, compound A cytotoxicity was assessed in bone marrow-derived macrophages with or without LPS or Pam3CSK4 for 16hrs and showed no cytotoxicity as can be seen in Figure EV2E (lines 311-313). This led us to the conclusion that in our preclinical models, compound A does not exhibit toxicities.

MCC950 is used as the best characterized NLRP3-inhibitory molecule in the field. Since it is not our proprietary compound, there are legal implications when running exploratory toxicity studies. However, in the inflammasome field, the presence of liver injury after MCC950 administration has been widely accepted. This is supported by the extra included reference Shah *et al* (2015, PMID: 26206150) reporting on the Phase II study conducted by Pfizer and presenting the

153 hepatotoxic liabilities attributed to MCC950. Mangan *et al* (2018, PMID: 30026524) mentioned
that MCC950's metabolically reactive furan moiety might underlie the observed toxicity, however
it remains speculation to date. To provide more clarity on this, additional references were added
to the text (line 139).

6. In supplementary fig 5, panel C has not been annotated.
We thank the reviewer for the observation. Supplementary fig 5C has been annotated and is now
converted into Appendix Figure S5C.

Referee #2 (Remarks for Author):

In this paper, Matico R et al. screened a series of compounds and identified compound A as a
novel chemical class of NLRP3 inhibitor, which possesses high selectivity and potency both in
vivo and in vitro. More impressively, the compounds deprived from the class inhibitors also
exhibited high efficacy in addressing CAPS mutations without inducing toxic side effects. Overall,
the findings are novel and interesting, providing priming candidates for the NLRP3-relevant
disorders, such as CAPS. However, the following concerns and questions should be addressed
to further strengthen the study.

Major comments:

1. The authors screened a novel chemical class cluster which could suppress NLRP3
inflammasome activation. Whether the class have a common feature, such as structural
similarity? The authors should explore or discuss it.

We thank the reviewer for this comment. Cluster 1 is significantly distinct from the diaryl
sulfonylurea-containing compounds as they don't contain a sulfonylurea core. The text was
edited to enhance the understanding of the reader (lines 191-192).

2. The authors showed the compound A-H inhibited NLRP3 activation in different degrees. Why
they exhibited the different role in the progress? Additionally, were compound A-H represented
the class inhibitors and screened out casually?

Within cluster 1, compound B was the original compound identified from the screening
campaign. As the properties of the molecule were not appropriate for further development, a
structure-activity relationship (SAR) campaign was conducted going from compound B and C to
E, F, G, H ending up with compound A. The SAR looked for compounds with improved clearance
and solubility, while maintaining a high potency and selectivity. This process is thoroughly
described in section 3 of the results (lines 240 – 285).

3. It has been reported a same chemical class of highly potent and selective NLRP3-targeted
inhibitors which reveals a well-defined molecular mechanism to complement existing MCC950-
based NLRP3 inhibitors in pharmacological studies (doi: 10.26508/lsa.202402644), and lots of
findings in the paper are similar to it, which greatly limits the innovation of the study.

We thank the reviewer for highlighting. Indeed, during the preparation of this manuscript,
compounds similar to compound C (in this manuscript) were described by Vande Walle *et al*
(2024), Velcicky *et al* (2024) and Li *et al* (2023), indicating that our screening funnel is robust.
However, in addition, we describe a successful SAR campaign to mitigate the liabilities
associated with this tricycle core which resulted in the identification of the bicyclic compounds
F, G, H ending up with compound A that had the desired properties and is chemically distinct from
the earlier mentioned molecules. We went on to characterize the chemically optimized
compound A with a full *in vitro* and *in vivo* assessment. This compound is chemically distinct and
not yet described and shows metabolic improvement potentially allowing its clinical
development. We strongly believe that our compound A is significantly different and represents
great importance to the field. We also filed patent applications for the bicyclic compounds.
Furthermore, this manuscript provides a better characterization of the physico, chemico and
pharmaco-kinetic properties of the compounds.

4. In Fig.3E, the inhibition of dose of Compound E in IL-1 β secretion seems almost indistinctive,
which was controversial with the conclusion of the authors.

We thank the reviewer for this observation and anticipate this is regarding Fig 3B instead of 3E.
Please note that the Y-ax is in Log scale. Compound E was able to significantly (50 mg/kg group:
* p = 0.0154, 12.5 mg/kg group: * p = 0.0491) reduce the release of IL-1 β in response to LPS as
as depicted by the * in the graph. MCC950 was used as positive control in this experiment the level
of IL-1 β that is NLRP3-dependent (MCC950 group: **** P < 0.0001) (lines 250 – 255). Statistical
tests and p-values were added to the figure legends.

5. In Fig.2 and Suppl Fig.2, why the authors used NLRP3- Δ PYD, but not the full-length NLRP3, to
explore the interaction of compound and NLRP3?

We observed that NLRP3- Δ PYD is more stable for biophysical and structural studies, likely due to
PYD-induced aggregation *in vitro*, which limited the experimental window and led to inconsistent
results. To accelerate structural studies in drug discovery, we used NLRP3 without PYD. Although
oligomerization differs, the NACHT domain structure remains highly conserved compared to full-
length NLRP3 in the closed/inhibited state.

6. MCC950 interacts the WALKER B motif of NLRP3, blocks ATP hydrolysis and inflammasome
formation, thereby inhibiting inflammasome activation (doi: 10.1038/s41589-019-0277-7). The
authors showed compound C and MCC950 both bind to the same pocket of NLRP3 but interact
with the protein differently. Whether compound C also blocks ATP hydrolysis and inflammasome
formation. Whether compound A bind to the pocket?

We thank the reviewer for this question. We show that compound C is able to block ATP hydrolysis
and NLRP3 inflammasome formation (Appendix Figure S6, and Fig.1G)

Since compounds A and C are from the same series, we opted for a more cost-effective and time-
efficient approach by conducting molecular dynamics simulations instead of additional cryo-EM
studies. These simulations indicated that compound A targets NLRP3 similarly to compound C.
We have added a new section discussing these simulation results. We appreciate the reviewer's
question and have included the relevant part here for easy reference:

**“Compound A targets the same NACHT binding pocket as MCC950 or compound C (lines**
**286-303).”**

7. Fig.1F has no data showing the NLRP3 inhibition of compound C and D, just their chemical
structural formula. Fig. 1G lacks the data related to IC50 for TNF- α and IL-6 secretion.

We thank the reviewer for this observation. The text was altered to indicate the representative
compound C and D structures in Fig. 1F and their effect on inflammasome activation in Fig. 1G
(lines 192 – 196). The data on IL-6 and TNF was added to figure Fig. 1G as requested.

8. The authors should simultaneously detect both original and reconstructive metabolic stability
and solubility of compound A in Fig.3D to highlight the role of further optimization.

We thank the reviewer for this comment. The SAR described in Figure 3 tells the story of how the
compounds were optimized looking at potency, solubility, metabolic stability and permeability.
The tricycle core was associated with low metabolic stability and solubility as described in Fig.
3A (lines 240 - 249) which led us to search for smaller heterocyclic replacements (Fig.3 C) (lines
256 – 274). The compounds F – G – H each improved specific parameters. However, it was
compound A that combined appropriate metabolic stability and solubility with permeability and
potency (Fig. 3D) (lines 275 – 285) sufficient to achieve POC *in vivo* and *in vitro* as witnessed in
our extensive characterization (Fig. 4 and Fig. 5).

9. Fig.4K lacked the related data to demonstrate the inhibition of compound A in IL-1 β secretion.

We thank the reviewer for this observation. The text (lines 338 - 342) has been reformulated to
explain better the complete inhibition of IL-1 β in blood collected 3 hrs after dosing indicative of
enough functional compound in circulation to prevent inflammasome activation. Alternatively, in
blood collected at 18 hrs post dosing, the inhibition was lost indicative of not enough compound
present in the blood at that timepoint.

10. The biological effects of compound A in other NLRP3-related disease model, such as DSS-
induced colitis, should be investigated

We thank the reviewer for his suggestion; however, we selected the LPS-mediated model to
investigate the effect of the compounds in NLRP3 activation for compound selection. DSS-
induced colitis model is not a specific model for NLRP3 activation due to crosstalk of multiple
pathways and furthermore multiple studies have shown contradictory results in this model
(PMID:33143375, 38280217). As a more relevant and specific disease model, we chose
Cryopyrin-associated Periodic Syndrome (CAPS) model. CAPS is caused by pathogenic
mutations in the *NLRP3* gene thereby making it the most relevant NLRP3-mediated disease
model. We introduced the Muckle-Wells Syndrome (MWS) mutation in a mouse model that was
already described (Vande Walle *et al* PNAS, 2019). In addition, human THP-1 cells with the same
MWS mutation were used and primary PBMCs from an FCAS and NOMID patient were obtained
to show the effect of compound A on a directly dysregulated NLRP3 inflammasome. Altogether,

we believe that these data effectively show the potential impact of compound A on a clinically
relevant NLRP3-mediated disease.

11. Suppl Fig.13G lacked the data related lung and spleens of in vivo experiment (Page13) .

We thank the reviewer for this observation. Supplemental Fig.13G now converted to Figure EV5G
and H. Representative images from lungs were added to Figure EV5H to show significant
reduction in the incidence of pulmonary thrombosis induced by NLRP3 mutation without
inducing any histopathological findings. Spleen from compound A treatment was histologically
normal, and showed no effect of the compound A and are added here below for reference (lines
382 – 386).

WT control

Vehicle Cre NLRP3 mutant

Cmpd A Cre NLRP3 mutant

**Figure 5: representative images of spleens treated with vehicle or compound A.**

Minor comments:

1. The name of inhibitor in figure legend (compound JNJ-666) of Fig.1D was not in line with the
name in Figure (compound B).

We thank the reviewer for this observation and have changed the figure legend for Fig.1D (line
791).

2. The abbreviations are usually used defined at the first use in the abstract as well as in the main
text, such as Poc or RHS in the 3rd part of Results.

We thank the reviewer for this observation and have thoroughly screened the manuscript for
abbreviations and added in full upon first use throughout the text.

3. In the whole part of Result, the author should make the conclusion in line with the
corresponding Figures, such as the findings related NLRP3-KO should be described as Fig.5A
behind the conclusion.

We thank the reviewer for this comment and have updated the paragraph on Fig.5A to provide
clear understanding of results for reader's (lines 348 – 352).

4. The final sentence in Result section is summary, but the authors added a reference (20).

We thank the reviewer for this observation and have removed the reference at this position (line
405).

5. There are numerous mistakes in the manuscript (especially in Results section). Please have
the text examined carefully.
We thank the reviewer for this observation. We have examined the manuscript carefully to correct
mistakes and restructure sentences to make the result section more easily understandable.

Referee #3 (Remarks for Author):

NLRP3 activity is recognized to cause or fuel a number of highly prevalent diseases including
metabolic diseases (diabetes, obesity), neurodegenerations, cardiovascular diseases, therefore
its pharmacological inhibition has a high therapeutic potential. In this manuscript, Matico et al.
described a novel family of compounds that specifically inhibits NLRP3 by direct targeting.
Chemical optimization of the initial compounds led to the development of compound A with
physico-chemico properties that allow in vivo efficiency with oral delivery in mouse, and ex vivo
efficiency in human cell lines and primary cells. These compounds target the same pocket as
MCC950 but with a different binding orientation. While usage of MCC950 and its derivatives in
clinic may be jeopardized by their hepatotoxicity, compoundA is apparently not hepatotoxic in
mice, which would constitute a major advantage. This study is well conduct in its methodology
and interpretation of the results. The main limitation is that 3 other independent studies recently
described the same family of compounds as novel NLRP3 inhibitor, which may reduce the novelty
of this study for publication in Embo Molecular Medicine. Nevertheless, this study is the most
advanced concerning compound optimization for in vivo use.

Major points :

- Compounds with similar structure, with identical mode of action and binding site on NLRP3 was
recently described by 3 other groups (PMID: 38519142, PMID: 37756547, PMID: 38175811).
Although it confirms the robustness of the results, it may decrease the novelty of this manuscript.
Nevertheless, this manuscript provides a better characterization of the physico, chemico and
pharmaco-kinetic properties of the compounds.

We thank the reviewer for highlighting this. Indeed, during the preparation of this manuscript,
compounds similar to compound C (in this manuscript) were described by Vande Walle *et al*
(2024), Velcicky *et al* (2024) and Li *et al* (2023), indicating that our screening funnel is robust. In
addition, we continued to describe a successful structure-activity relationship campaign
(section 2 of results) to mitigate the liabilities associated with this tricycle-containing compound
series. This ultimately resulted in the identification of the bicyclic-core compounds that showed
improved physico-chemical properties suitable for further development. We went on to
characterize the chemically optimized compound A with a full *in vitro* and *in vivo* assessment.
This compound is chemically distinct (and patented) and not yet described and shows metabolic
improvement allowing the clinical development. We therefore strongly believe that compound A
shows significant improvement over the compounds described by the other groups.

- Lack of hepatotoxicity of this novel class of compounds is presented as one of the major
advantages as compared to MCC950-derivatives. It would be critical to compare the 2 class of
compounds in parallel to prove this point. Concerning reference supporting the hepatotoxicity of
MCC950, the authors cite ref 20 (PMID: 37223529), which itself cites other reviews

(<https://doi.org/10.1021/cen-09807-cover> and PMID: 33255820). It would be better that the
authors cite the original study that relates original data supporting this claim. Alternatively, the
authors should show that MCC950 causes hepatotoxicity in the same experimental settings as in
suppl Fig13G for direct comparison, in order to demonstrate the superiority of compound A over
MCC950 described at the end of page15

We thank the reviewer for the observation. MCC950 is used as the best characterized NLRP3-
inhibitory reference molecule however, legal consequences can be attached to running
exploratory toxicity studies with this molecule. Nonetheless, in the inflammasome field, the
presence of liver injury after MCC950 administration has been widely accepted. This is supported
by the original reference Shah *et al* (2015, PMID: 26206150) reporting on the Phase II study
conducted by Pfizer and presenting the hepatotoxic liabilities attributed to MCC950. *Mangan et al*
(2018, PMID: 30026524) mentioned that MCC950's metabolically reactive furan moiety might
underlie the observed toxicity, however it remains speculation to date. Per recommendation of
the reviewer, we added the forementioned references to the text (line 137-139).

Minor points:
- Compound B structure should be disclosed to allow comparison with other publication that
recently characterized compounds of the same cluster (PMID: 38519142)

We thank the reviewer for this observation. The structure of compound B was added to the
manuscript in Fig. 1F. The compound B and the NIC-12 compound (PMID: 38519142) are both
pyrolo-triazine acetamide compounds however they show significant differences both on right-
hand and left-hand-side as can be seen from the structures below.

**Figure 6. Structure of compounds B described in this manuscript in comparison to NIC-12**
**from PMID 38519142.**

- Suppl Fig1: Description of the quantification methods, normalization and statistical analysis
would be helpful

We thank the reviewer for this comment. Statistical analysis is added to the figure legends and
raw data explaining the quantification and normalization is now provided for all main figures.

Supplemental Fig 1 (now Appendix Figure S1) shows a schematical representation of the
screening funnel. Most assays were described in Figure 1. Figure 1A shows the results from the
phenotypic cellular assay on-target for NLRP3 and off-target for NLRC4. Selection criteria are
described in the text (line 172-174). 'Compounds falling in the lower right quadrant (>60% NLRP3
and <30% NLRC4) were identified as confirmed hits (Fig. 1A).' Additional information on the triage
of the clusters was added to the text (lines 179 - 181).

Figure B, C and D show representative results for the dose response – on/off target selection using
 NLRP3 versus NLRC4 stimuli. Figure 1E shows representative results for the PBMC assay used to
 select the compounds. For these assays, the compounds were tested in 2 independent runs,
 each in duplicate, and MCC950 dose-response is used as positive control. No statistical analysis
 was conducted as this describes a triage funnel starting from around 1 million compounds, ending
 up with 1 cluster. MCC950 was used in each assay as benchmark and selection criteria were set
 based on the response of MCC950.

 - Fig2E: It would be helpful to represent cmpdC and MCC950 structural localisation in exactly the
 same orientation, or with superimposed images to facilitate the comparison.

Thank you for the reviewer’s suggestion. In Fig. 2E, both structures are shown in the same
 orientation. Since MCC950 and compound C have distinct binding poses within the same pocket,
 overlapping the structures will provide a clearer view of their interactions to NLRP3. We’ve
 included two pairwise overlap figures in the extended view figures for different purposes—one in
 Fig. EV1D and the other in Fig. EV4F. These comparisons are explained in text line 223-232, line
 397-402. Figures are added below for your reference.

 Fig.2E

Fig. EV4F

Fig. EV1D

- Statistical tests, number of independent experiments and replicates should be indicated for all
figures

We thank the reviewer for highlighting this. When used, statistical tests and p-values were added
to the figure legends. In addition, the number of independent repeats and technical repeats are
listed in the files with raw data as well as in the figure legends.

- Fig3 panels and legend: abbreviations should be limited to the most standard ones, and
indicated in the legend in full words. Units should be expressed with the standard abbreviations.
For example, mpk should be modified to mg/kg to comply to standard abbreviations and units

We thank the reviewer for this observation. We have carefully looked at the abbreviations and
indicated the remaining used abbreviations in the figure legend (lines 821 – 833) and other places
throughout the manuscript.

- Fig4A-B,D-J: unprocessed data (in addition to the displayed % Inhibition) should be added in
supplemental figures

We thank the reviewer for this observation. Raw data was provided with this resubmission for all
main figures.

- Fig5A and Supplemental Fig10: secretion of cytokines following LPS injection are highly dynamic
in time. Plasma TNF usually peaks around 1h post-LPS injection. A kinetics analysis of IL-1b, TNF,
IL6 at 1h, 2h, 4h is necessary to support the authors claims

We thank the reviewer for this comment and confirm the dynamic profile of the cytokines that are
used. We show the kinetics of IL-1 β , IL-6 and TNF- α after the treatment with LPS up to 8 hrs. The
4 hrs timepoint was selected as this is the peak for IL-1 β and to allow prevention of animal
suffering. We were not able to go beyond 8 hrs as the suffering became too strong and experiment
would go against the ethical guidelines for humane endpoint.

For TNF- α , the peak is indeed at 1 hrs post-treatment, however, the levels are still high enough to
see if there is a problem with NFkb-mediated cytokine production.

**Figure 7: Kinetic experiment in mice treated with 5 mg/kg to determine the optimal timepoint**
 **to look at the effect of our compounds on IL-1 β release into systemic circulation.**

- Page 14: the link between the screen methods published in ref 38, and the methods or results
 presented in the present manuscript is unclear. The authors may rephrase these few sentences
 to better explain the point they attempt to make.

We thank the reviewer for this observation. We rephrased the section in the discussion to clarify
 the message. The screening method used by the group in Oxford was able to identify some
 inhibitors of the inflammasome pathway rather than NLRP3 directly (lines 417 – 425).

16th Oct 2024

Dear Dr. Van Opdenbosch,

Thank you for the submission of your revised manuscript to EMBO Molecular Medicine. We have now heard back from the three referees who we asked to re-evaluate your manuscript. As you will see from the reports below, the referees are overall supporting publication of your manuscript. However, all referees raise important concerns that should be addressed in the next and final round of revision.

Acceptance or rejection of the manuscript will depend on the completeness of your responses included in the next, final version of the manuscript. For this reason, and to save you from any frustrations in the end, I would strongly advise against returning an incomplete revision.

1) Please address all the referee comments. The point regarding ASC specks raised by referee #1 and #2 should be addressed experimentally. Referee #2 point 2 should be discussed. Please implement Referee #3 suggestions regarding statistics. Referee #3 reiterated point regarding the direct comparison should be discussed and acknowledged as a limitation of the study in the main manuscript text as direct comparison of the hepatotoxicity of compound A and MCC950 has not been performed. Accordingly, conclusions about superiority of the compound A should be only made when there is direct experimental evidence from the study otherwise it should be toned down or omitted.

2) Authors: We note name discrepancies in our submission system and in the manuscript for all authors. Please make sure that author names match in our system and in the manuscript. Also, order of the authors in the manuscript should be the same as in our submission system. Please correct.

3) Figures: Please upload individual, high-resolution files in TIFF, EPS or PDF format for each EV figure.

4) In the main manuscript file, please do the following:

- Please address all comments suggested by our data editors listed below:

o Data availability: Please note that the specific URLs for PDB ID 9DH3, EMD-46855 datasets need to be provided in the data availability statement.

o Figure legends:

1. Please note that information related to n is missing in the legend of figure EV 1f.

2. Please note that n=2 in figures 1b-e; 5l; EV 4e.

3. Please note that the exact p values are not provided in the legends of figures 3b; 5a-b.

- Add up to 5 keywords.

- Rename "Conflict of interest" to "Disclosure and competing interests statement". We updated our journal's competing interests policy in January 2022 and request authors to consider both actual and perceived competing interests. Please review the policy <https://www.embopress.org/competing-interests> and update your competing interests if necessary.

- Author contributions: Please remove it from the manuscript and specify author contributions in our submission system. CRediT has replaced the traditional author contributions section because it offers a systematic machine-readable author contributions format that allows for more effective research assessment. Please use the free text boxes beneath each contributing author's name to add specific details on the author's contribution. More information is available in our guide to authors:

<https://www.embopress.org/page/journal/17574684/authorguide#authorshipguidelines>

- Thank you for including Reagent Tables. Please use the template you can download using the link below. More information on how to adhere to this format as well as downloadable templates (.docx) for the Reagents and Tools Table can be found in our author guidelines: <https://www.embopress.org/page/journal/17574684/authorguide#structuredmethods>

An example of a paper with Structured Methods can be found here:

<https://www.embopress.org/doi/full/10.1038/s44320-024-00037-6#sec-4>

- Indicate in legends number and nature of replicates and exact p= values, not a range, along with the statistical test used. To keep the figures "clear" some authors found providing an Appendix table Sx with all exact p-values preferable. You are welcome to do this if you want to.

- In data availability section please add URLs for or PDB ID 9DH3 and EMD-46855. Use the following format to report the accession number of your data:

[data type]: [full name of the resource] [accession number/identifier] ([doi or URL or identifiers.org/DATABASE:ACCESSION])

Please check "Author Guidelines" for more information.

<https://www.embopress.org/page/journal/17574684/authorguide#availabilityofpublishedmaterial>

- Please place references before figure legends.

5) Funding: Please disclose all sources of funding in our submission system and in the manuscript under "Acknowledgement".

6) Appendix: Please add page numbers and indicate them in the table of content.

7) Source data: Please upload one zip folder per figure.

8) Synopsis:

- Synopsis text: Please remove it from the manuscript and upload it as a separate .doc file.

- Synopsis image: Please resize it to 550 px-wide x (300-600) px-high and upload it as high-resolution jpeg file.
- Please check your synopsis text and image before submission with your revised manuscript. Please be aware that in the proof stage minor corrections only are allowed (e.g., typos).

9) As part of the EMBO Publications transparent editorial process initiative (see our Editorial at <http://embomolmed.embopress.org/content/2/9/329>), EMBO Molecular Medicine will publish online a Review Process File (RPF) to accompany accepted manuscripts. This file will be published in conjunction with your paper and will include the anonymous referee reports, your point-by-point response and all pertinent correspondence relating to the manuscript. Let us know whether you agree with the publication of the RPF and as here, if you want to remove or not any figures from it prior to publication. Please note that the Authors checklist will be published at the end of the RPF.

10) Please provide a point-by-point letter INCLUDING my comments as well as the reviewer's reports and your detailed responses (as Word file).

I look forward to reading a new revised version of your manuscript as soon as possible.

Yours sincerely,

Zeljko Durdevic

*** Instructions to submit your revised manuscript ***

- 1) a .docx formatted version of the manuscript text (including Figure legends and tables)
- 2) Separate figure files*
- 3) supplemental information as Expanded View and/or Appendix. Please carefully check the authors guidelines for formatting Expanded view and Appendix figures and tables at <https://www.embopress.org/page/journal/17574684/authorguide#expandedview>
- 4) a letter INCLUDING the reviewer's reports and your detailed responses to their comments (as Word file).
- 5) The paper explained: EMBO Molecular Medicine articles are accompanied by a summary of the articles to emphasize the major findings in the paper and their medical implications for the non-specialist reader. Please provide a draft summary of your article highlighting
 - the medical issue you are addressing,
 - the results obtained and
 - their clinical impact.

6) Author contributions: the contribution of every author must be detailed in a separate section.

7) EMBO Molecular Medicine now requires a complete author checklist (<https://www.embopress.org/page/journal/17574684/authorguide>) to be submitted with all revised manuscripts. Please use the checklist as guideline for the sort of information we need WITHIN the manuscript. The checklist should only be filled with page numbers where the information can be found. This is particularly important for animal reporting, antibody dilutions (missing) and exact values and n that should be indicated instead of a range.

8) Every published paper now includes a 'Synopsis' to further enhance discoverability. Synopses are displayed on the journal webpage and are freely accessible to all readers. They include a short stand first (maximum of 300 characters, including space) as well as 2-5 one sentence bullet points that summarise the paper. Please write the bullet points to summarise the key NEW findings. They should be designed to be complementary to the abstract - i.e. not repeat the same text. We encourage inclusion of key acronyms and quantitative information (maximum of 30 words / bullet point). Please use the passive voice. Please attach these in a separate file or send them by email, we will incorporate them accordingly.

You are also welcome to suggest a striking image or visual abstract to illustrate your article. If you do please provide a jpeg file 550 px-wide x 300-600px high.

9) A Conflict of Interest statement should be provided in the main text

10) Please note that we now mandate that all corresponding authors list an ORCID digital identifier. This takes <90 seconds to complete. We encourage all authors to supply an ORCID identifier, which will be linked to their name for unambiguous name identification.

Currently, our records indicate that the ORCID for your account is 0000-0001-6168-2971.

Link Not Available

11) Include a Reagents and Tools Table as part of the Methods section, which can be downloaded from our author guidelines (<https://www.embopress.org/page/journal/17574684/authorguide#structuredmethods>)

Photos 400-800 DPI

*Additional important information regarding figures and illustrations can be found at <https://bit.ly/EMBOPressFigurePreparationGuideline>. See also figure legend preparation guidelines: <https://www.embopress.org/page/journal/17574684/authorguide#figureformat>

***** Reviewer's comments *****

Referee #1 (Comments on Novelty/Model System for Author):

The revised manuscript is thorough and employs state-of-the-art experimental setup for identifying and characterizing new NLRP3 inhibitors.

Referee #1 (Remarks for Author):

The authors have put a significant effort into revising the manuscript. The revision is thorough and addressed most of the comments raised by this reviewer.

However, in comment no-4 (page-3), the authors' representation of ASC specs using 10X resolution (in inculcate) is not appealing. In particular, at this resolution, the possibility of picking up false positive cells (ASC positive THP1 cells) is high due to the small size of the THP1 cells. It is recommended to biochemically capture ASC oligomerization (either through native PAGE or through crosslinking the oligomers in cell lysates) to validate the effect of these new NLRP3 inhibitors.

Referee #2 (Remarks for Author):

The authors have addressed most of my concerns. However, the following concerns should be further addressed.

1. The data related to ASC specks is not clear and convincing enough, and should be further explored (refer to reported articles, such as DOI: 10.1038/nm.3806). Additionally, the method relevant to ASC speck is missing.
2. The effect of the Compounds on cell viability should be detected before exploring their roles in NLRP3 activation and IL-1 β release.

Referee #3 (Comments on Novelty/Model System for Author):

The authors used ANOVA tests. Have they validated the normality of the data ?

Means and SD (not SEM) would be more appropriate.

3 papers already described this class of compounds

Referee #3 (Remarks for Author):

In the revised version of the manuscript, Matico et al. answered most of my concerns adequately. Nevertheless, while the lack of hepatotoxicity of this novel class of compounds is presented as the major potential superiority over diaryl sulfonylurea-containing compounds, there is no direct comparison of the hepatotoxicity of the 2 classes of compounds in the same settings. The authors should at least compare the 2 classes of compounds side by side in the used murine model.

We thank all the referees again for carefully examining our rebuttal manuscript. We appreciate their time in carefully reading the rebuttal and suggesting improvement. Please find our responses below highlighted in blue.

Referee #1 (Comments on Novelty/Model System for Author):

The revised manuscript is thorough and employs state-of-the-art experimental setup for identifying and characterizing new NLRP3 inhibitors.

Referee #1 (Remarks for Author):

The authors have put a significant effort into revising the manuscript. The revision is thorough and addressed most of the comments raised by this reviewer. However, in comment no-4 (page-3), the authors' representation of ASC specs using 10X resolution (in inculcate) is not appealing. In particular, at this resolution, the possibility of picking up false positive cells (ASC positive THP1 cells) is high due to the small size of the THP1 cells. It is recommended to biochemically capture ASC oligomerization (either through native PAGE or through crosslinking the oligomers in cell lysates) to validate the effect of these new NLRP3 inhibitors.

We thank referee 1 for thoroughly reading and acknowledging our efforts to improve the manuscript. We carefully examined referee's concern and now have provided biochemical/immunofluorescence assay to determine ASC speck in our study. Please see the new Fig. 4F and EV2F. We have performed an immunofluorescent staining and quantified for ASC specks upon NLRP3 activation in presence of compd A (new Fig 4F). Moreover, ASC speck formation upon NLRC4 activation was unaltered in presence of compd A (see below Figure 1 and 2). Furthermore, biochemical ASC-crosslinking assay in BMDM (Fig EV2F) showed inhibition of ASC speck formation upon NLRP3 activation by compd A (see below Figure 3). Original Figure 4F using THP-1 ASC mCherry cells was added to Appendix Fig S7. Main text was altered (lines 309-317, lines 660-677) accordingly. The ASC-mCherry figure of THP-1 is now added to appendix Fig. S7. Altogether, we strongly believe that now these results will satisfy referee's remaining concern.

Figure 1: quantification of ASC specks from immunofluorescent staining of BMDMs treated with MCC950 and Comp A followed by stimulation with LPS + Nigericin (NLRP3) and FlaTox (NLRC4).

Figure 2: representative images showing ASC specks (green) on immunofluorescent staining of BMDMs treated with MCC950 and Comp A followed by stimulation with LPS + Nigericin (NLRP3) and FlaTox (NLRC4).

Figure 3: biochemical analysis of ASC specks in BMDMs treated with MCC950 and Comp A followed by stimulation with LPS + Nigericin (NLRP3) and FlaTox (NLRC4).

Referee #2 (Remarks for Author):

The authors have addressed most of my concerns. However, the following concerns should be further addressed.

1. The data related to ASC specks is not clear and convincing enough, and should be further explored (refer to reported articles, such as DOI: 10.1038/nm.3806). Additionally, the method relevant to ASC speck is missing.

We thank referee 2 for examining our reading and acknowledging our rebuttal manuscript. We carefully examined referee's concern and now have provided biochemical/immunofluorescence assay to determine ASC speck in our study. Please see above answer to referee 1 for new analysis and data. We strongly believe that now these results will satisfy referee's remaining concern.

2. The effect of the Compounds on cell viability should be detected before exploring their roles in NLRP3 activation and IL-1 β release.

Thank you for this suggestion: the screening assay using the J774A.1 cells using Sytox green is a gain-of-signal assay and was designed to pick up inhibitors of the NLRP3 inflammasome and at the same time filter out cytotoxic molecules. The representative figure below shows a non-cytotoxic (blue) and cytotoxic (purple) control that was included in the screening. The higher concentrations of the purple compound were cytotoxic so the % activity falls within the noise of the assay and therefore this compound was not picked up as a hit from the screening.

This was added to the main text to make it more clear for the reader (line 157 - 159).

Figure 4: J774A.1 sytox green screening assay showing a non-toxic on-target (blue) and a toxic (purple) compound in dose response.

In addition, the final compound A was tested on its cytotoxic capacity which was addressed in lines 299-301.

Referee #3 (Comments on Novelty/Model System for Author):

The authors used ANOVA tests. Have they validated the normality of the data?

We thank the reviewer for this question. Cytokine data are widely accepted as using a log normal distribution. As we are showing log-transformed data, the normality assumption was fulfilled.

Means and SD (not SEM) would be more appropriate.

We thank the reviewer for this observation. We consulted our statisticians in support of this revision and confirm that the *in vitro* data needs to show SD instead of SEM. We have now changed SEM to SD in *in vitro* figures. As in the *in vivo* experiments, we are looking to compare the mean values between treatment, we feel that the SEM is appropriate here.

3 papers already described this class of compounds
We indeed agree with referee 3 that the tricycle class of compounds has already been described thereby underlying the success of our phenotypic screening cascade. However, we significantly changed and improved the molecule ending up with a bicyclic phthalazine that is suitable for further development with enhanced stability and solubility. Therefore, we strongly believe that because to this change in chemistry, we are in novel space with potential clinical candidate for NLRP3 inhibition. These novel structures have also been patented (WO2022229315).

Referee #3 (Remarks for Author):

In the revised version of the manuscript, Matico et al. answered most of my concerns adequately. Nevertheless, while the lack of hepatotoxicity of this novel class of compounds is presented as the major potential superiority over diaryl sulfonylurea-containing compounds, there is no direct comparison of the hepatotoxicity of the 2 classes of compounds in the same settings. The authors should at least compare the 2 classes of compounds side by side in the used murine model.

We thank the reviewer for this observation. The statements of superiority on hepatotoxicity have been toned down or omitted from the main text and a conclusion is added to the discussion that a direct comparison is required to fully understand the effect of both compounds on hepatotoxicity to allow the further clinical development of compound A (lines 466-471).

26th Nov 2024

Dear Dr. Van Opdenbosch,

We are pleased to inform you that your manuscript is accepted for publication and is now being sent to our publisher to be included in the next available issue of EMBO Molecular Medicine.
